# CONSTRUCTIVE DISTORTION: IMPROVING MLLMS WITH ATTENTION-GUIDED IMAGE WARPING

**Dwip Dalal**[1], **Gautam Vashishtha**[2], **Utkarsh Mishra**[3], **Jeonghwan Kim**[1],
**Madhav Kanda**[1], **Hyeonjeong Ha**[1], **Svetlana Lazebnik**[1], **Heng Ji**[1], **Unnat Jain**[4]

[1] University of Illinois Urbana–Champaign   [2] Skan AI   [3] Texas A&M University
[4] University of California, Irvine

dwip2@illinois.edu

## ABSTRACT

Multimodal large language models (MLLMs) often miss small details and spatial relations in cluttered scenes, leading to errors in fine-grained perceptual grounding. We introduce `AttWarp`, a lightweight method that allocates more resolution to query-relevant content while compressing less informative areas, all while preserving global context. At test time, AttWarp closes a simple self-correction loop: the MLLM first produces cross-modal attention on the original image, which we use to rectilinearly warp the input and re-run the same frozen model, reallocating resolution toward regions it deems important without changing weights or architecture. This attention-guided warping preserves all original image information but redistributes it non-uniformly, so small objects and subtle relationships become easier for the same model to read while the global layout remains intact. Across nine benchmarks (TextVQA, GQA, DocVQA, POPE, MMMU, MIA-Bench, MMVP, RealWorldQA, BLINK) and four MLLMs (LLaVA, Qwen-VL, InternVL, and InstructBLIP), `AttWarp` consistently improves accuracy, strengthens compositional reasoning, and reduces hallucinations, outperforming four competitive baselines that manipulate raw images at test time. Together, these results show that attention-guided warping prioritizes information relevant to the query while preserving context, and that the same MLLMs perform better when given such warped inputs. The code and demos are available on the project page: https://dwipddalal.github.io/Attwarp/

## 1 INTRODUCTION

Humans perceive certain regions of a scene by dynamically allocating high-resolution resources to areas of interest. This behavior is described as the interplay between *foveal vision*, which provides detailed perception at the center of attention, and *peripheral vision*, which rapidly scans the broader scene in lower resolution (Carrasco, 2011). This warped way of perceiving our surroundings is dynamic and dependent on task demands. As Gibson argued (Gibson, 1966), perceptual systems actively restructure their input, sampling dense information where it's most needed. This introduces a form of distortion, not to obscure, but to enhance relevance.

While advanced deep learning models incorporate some aspect of this through attention mechanisms, they leave significant issues in fine-grained perceptual grounding. Multimodal LLMs often fail to identify small details, distinguish between similar objects, and understand complex spatial relationships in cluttered scenes, leading to misclassification and incorrect reasoning (Yang et al., 2024b; He et al., 2025; Kim & Ji, 2024). In this work, we investigate the benefits of this principle of warped perception in the context of modern multimodal LLMs. Particularly, we investigate the research questions: *what is an effective method for warping images that preserve global context while expanding task-relevant regions? Would existing MLLMs perform better with warped images?*

To answer the first research question, we devise a lightweight recipe for warping images that preserves global context while expanding task-relevant regions. Our method, Attention-Guided Image Warping (`AttWarp`), operates as a plug-and-play enhancement, requiring no modifications to the underlying

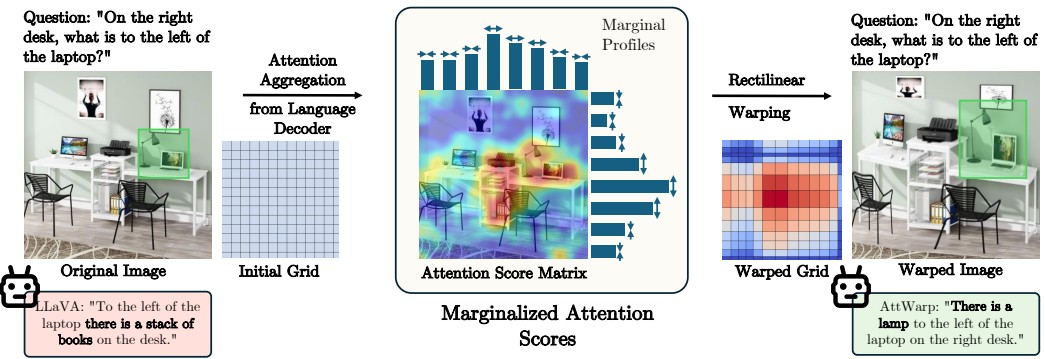

**Figure 1: `AttWarp` overview.** Given a query, our method extracts cross-modal attention maps from the MLLM's language decoder, aggregates them into marginal attention profiles, and uses rectilinear warping to expand high-attention regions while compressing low-attention areas. The warped image is then processed by the same MLLM, which now produces the correct answer.

MLLM architecture. As illustrated in Fig. 1, given an input image and a query, we first extract cross-modal attention maps from the MLLM's language decoder. These attention maps are aggregated into an *Attention Score Matrix* and further condensed into 1D *marginal attention profiles* along both horizontal and vertical axes (Fig. 1, middle). These profiles quantify the importance of each row and column in the image, with taller bars indicating regions that should receive more visual real estate. These marginal profiles guide our novel *rectilinear warping* process, which non-uniformly resamples the image grid, expanding high-attention regions and compressing low-attention areas, as shown by the red-to-blue gradient in Fig. 1 (right). Crucially, our choice of warping ensures that all original image information is preserved, maintaining global context, unlike methods that crop or mask parts of the image. In the warped image, task-relevant objects such as the lamp and laptop highlighted by the green bounding box are visually enlarged, making fine-grained details and spatial relationships more accessible to the MLLM. Towards the second research question, we find that such warped images, when processed by the same MLLM, lead to improved performance across nice multimodal benchmarks, and the idea generalizes to multiple MLLM backbones. We empirically validate that our rectilinear design is crucial to this improvement without changing a single parameter of the MLLM. We extend this framework in two directions: repeating this attention-guided warp as an iterative self-correction step yields further improvements(`AttWarp-Chain`), and we learn a distilled model to directly predict marginal profiles instead of estimating them from MLLM attention maps (`AttWarp-Distill`).

Conceptually, `AttWarp` utilizes attention to modify the input image itself, different from the typical use of attention to reweight latent features. Importantly, our contribution is complementary to research that improves attention mechanisms in MLLMs, such as refining attention heads (Bi et al., 2024; Kang et al., 2025), adding auxiliary objectives (Yan et al., 2024), or redesigning cross-modal fusion layers (He et al., 2025). Finally, note that we intervene *before* feature extraction, while the above methods operate *after* the image has already been encoded, often from features that have already lost critical spatial detail (Pantazopoulos et al., 2024).

In summary, our key contributions are: 1) A *lightweight method* `AttWarp` that addresses the fine-grained perceptual grounding challenges identified in multimodal LLMs by modifying input images before feature extraction, requiring no MLLM finetuning. 2) Consistent empirical gains over *four competitive baselines*, across *nine standard vision-language benchmarks* that test diverse capabilities such as fine-grained multimodal understanding, compositional reasoning, and hallucination mitigation. 3) Demonstrating generalization across *multiple MLLM backbones* and attention sources, underscoring its plug-and-play compatibility. 4) *Rigorous analysis* validating that our warps indeed expand task-relevant regions and `AttWarp`'s rectilinear design helps preserve the original data distribution.

## 2 RELATED WORK

**Perception challenges in MLLMs.** MLLMs such as Flamingo (Alayrac et al., 2022), LLaVA(Liu et al., 2023b), Qwen (Yang et al., 2024a), MiniGPT-4 (Zhu et al., 2023), and GPT-4V (OpenAI, 2023)

have advanced image-grounded dialogue and reasoning (Mitra et al., 2024; Dong et al., 2024; Zhang et al., 2025a; 2024b). Yet they still struggle with *fine-grained* perception—missing small attributes (e.g. object (Zhang et al., 2024a)), misclassifying sub-categories (Geigle et al., 2024; Kim & Ji, 2024; Yu et al., 2025), and confusing geometric primitives (Zhang et al., 2025b). These limitations motivate our work, which aims to enhance the fine-grained perception of the MLLMs by improving query-specific spatial resolution prior to feature extraction.

**Different Approaches for Fine-Grained Visual Understanding.** We group prior efforts into the following categories: (i) Bounding-box methods (Peng et al., 2023; Chen et al., 2023a; Zhang et al.; Lu et al., 2024) steer attention by feeding cropped regions obtained from bounding boxes. (ii) Mask-based methods (Chen et al., 2024b; Yuan et al., 2024; You et al., 2023; Zhang et al., 2024b) supply pixel-accurate masks—often from Segment-Anything (Kirillov et al., 2023). (iii) Cascade methods (He et al., 2025; Yang et al., 2023a; Yu et al., 2024) overlay detector cues or saliency heatmaps to bias the input. (iv) Reasoning methods (Surís et al., 2023; Wei et al., 2022) decompose queries into low-level visual steps. Our approach achieves stronger grounding while avoiding extra detectors, masks, or multi-step reasoning chains.

**Pixel-Level Warping Techniques for Saliency Emphasis.** Classical work includes seam carving (Rubinstein et al., 2010), saliency-aware warps (Wolf et al., 2007; Recasens et al., 2018), energy minimisation (Karni et al., 2009), finite-element grids (Kaufmann et al., 2013), mesh parametrisation (Guo et al., 2009), and seam & scale methods (Zhang et al., 2009). Learning-based variants explore adaptive resizing (Talebi & Milanfar, 2021), saliency enhancement (Ghosh et al., 2019; Miangoleh et al., 2023), and domain adaptation (Zheng et al., 2025), with contemporaneous magnification work (Mao et al., 2025). Many existing approaches, such as those employing energy minimization or seam carving, are optimization-based. Consequently, processing each input sample can take several minutes. In contrast, our proposed method leverages a single forward pass of a Cumulative Distribution Function (CDF), enabling near-instantaneous processing. Here, we build on saliency-aware sampling by introducing a *query-conditioned, rectilinear* warp that preserves the image's regular grid structure, ensuring compatibility with the MLLM's vision encoder.

## 3 ATTENTION-GUIDED IMAGE WARPING

We propose a simple and effective test-time technique to improve visual grounding of MLLMs. Instead of feeding the original image directly, we apply a spatial transformation guided by the model's internal attention, reshaping image regions based on their relevance to the query. Below, we provide a high-level overview of `AttWarp`, followed by a detailed description of its components.

**Overview.** As illustrated in Fig. 1, `AttWarp` uses cross-modal attention maps from deeper layers of the MLLM (see Sec. 3.4) to guide a distribution-preserving non-uniform resampling of the original image. This resampling operation, termed **rectilinear warping** (Sec. 3.1), redistributes pixel density across the image: regions with high attention are spatially expanded, while less relevant regions are compressed. Relevance is always defined with respect to the specific query, making the warping adaptive to task semantics. Crucially, the warped image retains a regular grid structure, ensuring compatibility with standard vision encoders. Next, in Sec. 3.2, we introduce `AttWarp-Chain` an extension of `AttWarp` that iteratively applies multiple warps, grounding each step in the model's evolving attention and improving performance on complex queries. Finally, in Sec. 3.3 we introduce `AttWarp-Distill`, a computationally efficient version optimized for inference speed, which runs $3\times$ faster than prior methods by shifting additional computation to training time through learned warping functions.

### 3.1 RECTILINEAR IMAGE WARPING

Given an input image $\mathbf{I} \in \mathbb{R}^{H \times W \times 3}$ and an attention score matrix $A \in \mathbb{R}^{H \times W}$ (from Sec. 3.4), our goal is to obtain a function $F$ that transforms $\mathbf{I}$ to a warped image $\mathbf{W} = F(\mathbf{I}; A)$. The warping function $F$ is designed to magnify important regions (high attention) and compress less relevant ones.

First, we compute marginal attention profiles (PDFs) along rows and columns to decompose the 2D attention matrix into 1D score vectors:

$$\text{Horizontal Attention Profile: } m_x(j) = \sum_{i=1}^{H} A_{ij}, \text{ Vertical Attention Profile: } m_y(i) = \sum_{j=1}^{W} A_{ij}. \quad (1)$$

Here, $i \in (1, 2 \ldots H)$ and $j \in (1, 2 \ldots W)$. This decomposition facilitates rectilinear warping, enabling independent transformations along the horizontal and vertical axes while preserving the grid structure. Subsequently, we convert these marginals into cumulative distribution functions (CDFs):

$$M_x(j) = \frac{\sum_{k=1}^{j} m_x(k)}{\sum_{k=1}^{W} m_x(k)}, \qquad M_y(i) = \frac{\sum_{k=1}^{i} m_y(k)}{\sum_{k=1}^{H} m_y(k)}. \tag{2}$$

These resulting cumulative functions $(M_x, M_y)$ are monotonically increasing and therefore invertible. We define the warping functions using their inverses, known as the *Inverse Distribution Functions*:

$$f_X^{\text{Warp}}(j) = W \cdot M_x^{-1}(j/W), \quad f_Y^{\text{Warp}}(i) = H \cdot M_x^{-1}(j/H) \tag{3}$$

where $i \in (1, 2 \ldots H)$ and $j \in (1, 2 \ldots W)$. Together, these inverse mappings $f_X^{\text{Warp}}$ and $f_Y^{\text{Warp}}$ constitute the overall warping transformation $F$, yielding the warped image. The final warped image is computed through bilinear sampling, applied along all three channels:

$$\mathbf{W}[i, j] = \texttt{Interpolate}(\mathbf{I}, \texttt{Bilinear})(f_Y^{\text{Warp}}(i), f_X^{\text{Warp}}(j)). \tag{4}$$

## 3.2 ITERATIVE IMAGE WARPING (ATTWARP-CHAIN)

The optimal degree of warping depends on the query and the image. Queries focusing on small details benefits from strong warping, while broader scene may require minimal warping. A naive way is to use a superlinear (or sublinear) transformation over the attention score matrix to upweigh (or downweigh) the attention-guided warp. In this section, we introduce a more intuitive and nuanced scheme that performs better called `AttWarp-Chain`.

We build an iterative warping method based on an empirical observation i.e., warping improves MLLMs attention (See 4.3) and enhanced attention maps subsequently yield better warping. Leveraging this insight, we develop `AttWarp-Chain` which after each iteration, extracts an updated attention map, and progressively refines the warp applied in the previous iteration. Formally,

$$\text{Initialization: } \mathbf{W}^{(0)} = \mathbf{I}, \quad \text{Chain step: } \mathbf{W}^{(d)} = F(\mathbf{W}^{(d-1)}; A^{(d-1)}). \tag{5}$$

Here, $A^{(d-1)}$ denotes the attention map computed from the warped visual input $\mathbf{W}^{(d-1)}$.

A practical question left to answer is when to terminate the chain of iterative warping steps? Instead of encoding this as a hyperparameter, we propose a more adaptive route. As the relevant query-specific region expands, the attention map spreads more uniformly over the image. Eventually, the attention distribution stabilizes, indicated by minimal changes between successive attention maps. We quantify this stability through the following stopping criterion:

$$\mathcal{D}_{\text{KL}}\left(P^{(d)}|P^{(d-1)}\right) < \epsilon_{\text{KL}}, \tag{6}$$

where $P^{(d)}$ and $P^{(d-1)}$ are normalized attention probability distributions from iterations $d$ and $d-1$, respectively. This termination ensures `AttWarp-Chain` achieves optimal spatial emphasis while mitigating the risks associated with noisy or overly aggressive warping. We quantitatively show effectiveness of `AttWarp-Chain` and termination criteria in App. D.3.

## 3.3 LEARNING TO PREDICT MARGINAL ATTENTION PROFILES (ATTWARP-DISTILL)

Many applications (e.g., edge AR and embodied agents) need fast, precise grounding. Masking, cropping, or re-running attention adds latency; even `AttWarp-Chain` requires multiple passes. We therefore learn a *single-pass* predictor that outputs the horizontal and vertical marginals, $m_x(j)$ and $m_y(i)$, directly from an image–text pair. This neural functional approximation removes attention retrieval at inference and keeps the warping semantics of Sec. 3.1.

**Teacher: generating marginal targets.** We use the base MLLM and the attention-extraction pipeline from Sec. 3.4 to produce training targets. For each image–text pair $(\mathbf{I}, \text{text})$ we compute the attention score matrix $A$, derive axis-wise marginals via equation 1, and normalize them to unit mass, yielding $(m_x, m_y)$. This defines the dataset $\mathcal{D} = \{(\mathbf{I}_n, \text{text}_n, m_{x,n}, m_{y,n})\}_{n=1}^{N}$. Constructing these targets is intentionally done once offline to train the student; at inference the student replaces this pipeline, amortizing cost and enabling single-pass usage.

**Student: `AttWarp-Distill`.** Using these offline targets, we train a compact network to predict $(m_x, m_y)$ from an image–text pair (architecture in Fig. 2). We encode the image with CLIP ViT-L/14 to obtain vision tokens $\mathbf{Z}$ and obtain text tokens $\mathbf{q}$ from a tokenizer $E_t$ applied to the query. Text conditions the vision tokens through Feature-wise Linear Modulation (FiLM) (Perez et al., 2018). A small MLP maps the text tokens $\mathbf{q}$ to per-channel scale and shift parameters $(a, b)$, and applies them channel-wise to obtain the modulated tokens $\tilde{\mathbf{Z}} = a \odot \mathbf{Z} + b$. We then upsample $\tilde{\mathbf{Z}}$ to $(H, W)$ and average along one axis at a time to obtain two 1D summaries (horizontal and vertical). Two light Conv1D heads turn these summaries into logits, which a SoftMax converts to valid marginals $(\hat{m}_x, \hat{m}_y)$. Training minimizes the expected $L_1$ discrepancy over $\mathcal{D}$, i.e., the average of $\|\hat{m}_x - m_x\|_1 + \|\hat{m}_y - m_y\|_1$ across samples.

**Single-pass, fast inference.** At inference, given $(\mathbf{I}, \text{text})$, `AttWarp-Distill` outputs $(\hat{m}_x, \hat{m}_y)$ in one forward pass. We convert them to CDFs $(\hat{M}_x, \hat{M}_y)$ via Eq. equation 2, invert to coordinates using Eq. equation 3, and bilinearly sample the image as in Eq. equation 4 to obtain $\mathbf{W}$. This retains the semantics of Sec. 3.1 while reducing cost. Training details appear in App. G.3.

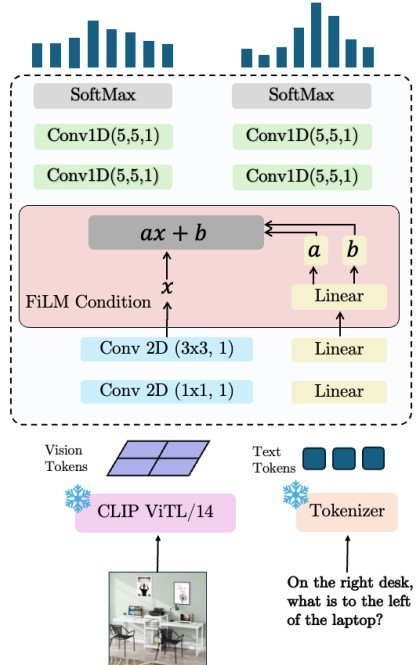

**Figure 2:** `AttWarp-Distill` architecture: CLIP vision tokens are FiLM-modulated by text and projected to 1D marginal predictors.

### 3.4 ATTENTION SCORE MATRIX IMPLEMENTATION

Here, we describe a general recipe for constructing the attention score matrix $A$ for any base MLLM. The procedure has two steps: (1) *Attention retrieval*, which reads raw cross-attention weights, and (2) *Attention aggregation*, which collapses those weights into a single spatial map for the given image–text pair.

**Attention map retrieval.** The image $\mathbf{I}$ is tokenized into $n_{\text{img}} = h_{\text{feat}} w_{\text{feat}}$ vision tokens on a $h_{\text{feat}} \times w_{\text{feat}}$ grid. After processing $\mathbf{I}$ and the text, the MLLM produces $n_{\text{out}}$ output tokens. From selected decoder layers $\mathcal{L}$ and all attention heads $(n_{\text{heads}})$, we obtain cross-attention matrices $a^{(\ell, h)} \in \mathbb{R}^{n_{\text{out}} \times n_{\text{img}}}$; entry $a_{m,t}^{(\ell, h)}$ is the weight from output token $m$ to image token $t$.

**Attention map aggregation.** We now average over output tokens, heads, and the chosen layers to form a single spatial map. Each image token index $t \in \{1, \ldots, n_{\text{img}}\}$ corresponds to grid location $(i, j)$ via $t = (i-1) w_{\text{feat}} + j$. The aggregated score at $(i, j)$ is

$$\tilde{A}_{i,j} = \frac{1}{n_{\text{out}} \cdot n_{\text{heads}} \cdot |\mathcal{L}|} \sum_{\ell \in \mathcal{L}} \sum_{m=1}^{n_{\text{out}}} \sum_{h=1}^{n_{\text{heads}}} a_{m,t}^{(\ell, h)}. \tag{7}$$

Here, $a_{m,t}^{(\ell, h)}$ denotes the weight from output token $m$ to image token $t$. We upsample $\tilde{A}$ from $h_{\text{feat}} \times w_{\text{feat}}$ to the image resolution $H \times W$ using Lanczos, smooth with a $k \times k$ AvgPool, and optionally apply a scalar transform $\mathcal{T}$ to control sharpness. The final attention score matrix is $\mathcal{T}(\tilde{A}_{ij}), \mathcal{T} \in \{x, x^2, \sqrt{x} \ldots\}$. Here, $i \in \{1, 2, \ldots, H\}$, $j \in \{1, 2, \ldots, W\}$. Sharper transforms like $x^2$ emphasize high-attention regions more than linear ones, which is useful for fine-grained queries. For LLaVA, we choose the 20th layer *i.e.* $\mathcal{L} = \{20\}$. For Qwen, we use the 16th layer *i.e.* $\mathcal{L} = \{16\}$. The strategy for choosing layer(s) $\mathcal{L}$ and attention aggregation method is explained quantitatively in App. B. For more details on implementation and design choices, refer App. G.1.

## 4 EXPERIMENTS

In this section, we detail our experimental framework and findings. We begin by describing the evaluation benchmarks and the baseline models used for comparison (Sec. 4.1). Next, we present our key quantitative results across multiple multimodal benchmarks (Sec. 4.2). Finally, we include ablations and analyses to share insights into `AttWarp` (Sec. 4.3)

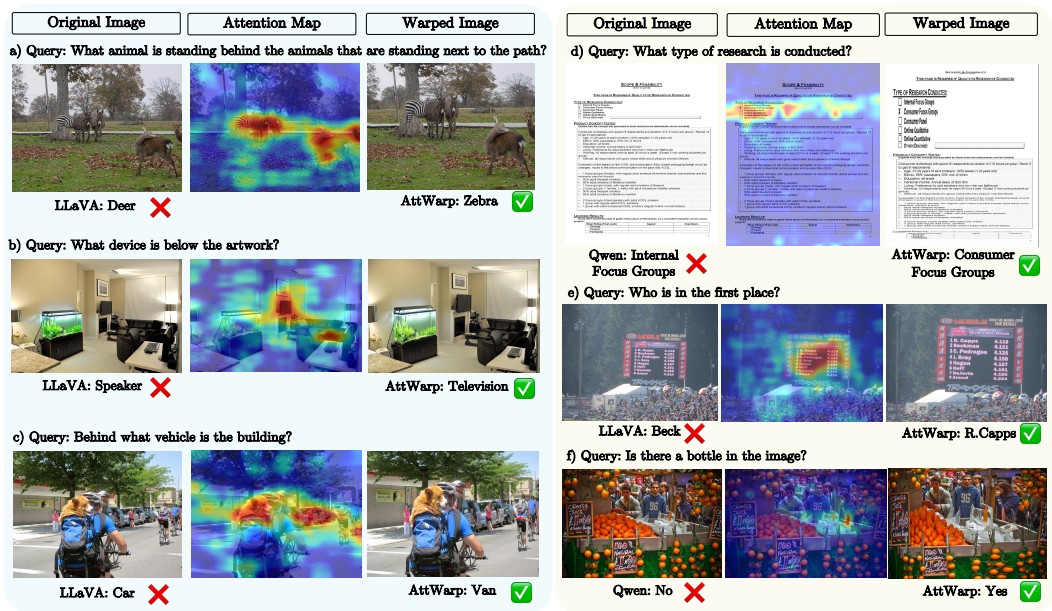

**Figure 3:** AttWarp improves compositional & spatial reasoning *e.g.* from GQA dataset (a) correctly identifying zebra behind the path, (b) television below the artwork; text understanding in documents *e.g.* from DocVQA (d) consumer focus groups, fine-grained recognition of small/occluded objects *e.g.* from POPE (f) detecting a bottle

## 4.1 BENCHMARKS AND BASELINES

**Benchmarks.** We evaluate `AttWarp` on *nine diverse benchmarks* designed to assess key multimodal capabilities, including general multimodal understanding, compositional reasoning, spatial relationships, visual hallucination, and fine-grained visual understanding (Fig. 3 shows qualitative results).
• *GQA (visual reasoning):* Reasoning about objects, attributes, and relations in real-world images (Hudson & Manning, 2019).
• *TextVQA (scene text understanding):* Answering questions that require reading and grounding text in natural images (Singh et al., 2019).
• *DocVQA (document image understanding):* Extracting and reasoning over textual and structural information in scanned documents (Mathew et al., 2021).
• *POPE (robustness evaluation):* Probing fine-grained hallucination and reliability in vision–language models (Li et al., 2023).
• *MMMU (general multimodal understanding):* Broad multi-disciplinary evaluation across STEM, humanities, and social sciences (Yue et al., 2024).
• *RealWorldQA (spatial reasoning in the wild):* Spatial reasoning and relative localization in complex real-world scenes captured from vehicles and other environments (xAI, 2024).
• *BLINK (fine-grained perception):* Core visual perception skills via fine-grained tasks (e.g., relative depth estimation, object localization, counting) cast into a VQA-style interface (Fu et al., 2024).
• *MMVP (fine-grained perception):* Evaluation on *CLIP-blind* image pairs that stress robustness to systematic changes in visual patterns such as orientation and viewpoint (Tong et al., 2024).
• *MIA-Bench (instruction following):* Multimodal instruction following under layered constraints on style, length, and content while remaining visually grounded (Qian et al., 2024).
We report the result of `AttWarp` on GQA, TextVQA, DocVQA, POPE, and MMMU in Tab. 1 and results on MMVP, BLINK, RealWorldQA, and MIA in Tab. 2.

**Baselines.** For a rigorous evaluation, beyond the base MLLMs, we also compare `AttWarp` to *four representative baselines for test-time visual intervention*. Baselines span strategies for editing input image to guide MLLM attention directly at inference (see Fig. 4 for visual examples).

• `FGVP` (Yang et al., 2023b) (Region Isolation via Masking): applies semantic masks on the target region and reversely blurs the background (or applies a green mask) outside the target region.
• `SoM` (Yang et al., 2023a) (Visual Grounding with Explicit Markers): segments the input image semantically using an off-the-shelf model, and labels each segment with a unique visual marker.

**Table 1:** Main results on TextVQA, GQA, MMMU, POPE, and DocVQA datasets in accuracy (%). The Δ Accuracy row reports the absolute improvement of `AttWarp-Chain` over the base MLLM.

| # | Methods | Key Technique | TextVQA | GQA | MMMU | POPE | DocVQA |
|---|---------|---------------|---------|-----|------|------|--------|
| | LLaVA (Liu et al., 2024a) *(MLP vision-language connector & open data)* | | | | | | |
| 1 | Base MLLM | | 49.3 | 60.5 | 36.9 | 85.3 | 18.1 |
| 2 | FGVP-mask (Yang et al., 2023b) | Green mask overlay | 39.4 | 59.2 | 36.1 | 85.3 | 19.0 |
| 3 | FGVP-blur (Yang et al., 2023b) | Blur background | 33.9 | 59.5 | 35.0 | 83.1 | 18.6 |
| 4 | SoM (Yang et al., 2023a) | Grounded segments | 18.8 | 54.5 | 35.6 | 78.5 | 15.8 |
| 5 | API (Yu et al., 2024) | Alpha channel fade | 49.9 | 60.6 | 36.9 | 85.9 | 17.4 |
| 6 | ViCrop (Zhang et al., 2025a) | Add object crop | *56.3* | *60.9* | *37.2* | *87.0* | *22.5* |
| 7 | AttWarp *(ours)* | Rectilinear warping | 58.1 | 63.7 | 40.4 | 87.5 | 25.5 |
| 8 | AttWarp-Distill *(ours)* | Efficient inference | 57.2 | 62.7 | 38.8 | 87.4 | 22.4 |
| 9 | AttWarp-Chain *(ours)* | Adaptive Chains | **60.3** | **64.4** | **41.6** | **88.2** | **27.6** |
| 10 | Δ Accuracy | | +11.0 | +3.9 | +4.7 | +2.9 | +9.5 |
| | Qwen (Yang et al., 2024a) *(Cross-attention VL adapter & partially closed data)* | | | | | | |
| 11 | Base MLLM | | 81.0 | *62.4* | 47.3 | 86.1 | 77.3 |
| 12 | FGVP-mask (Yang et al., 2023b) | Green mask overlay | 77.3 | 55.8 | 46.0 | 84.4 | 56.6 |
| 13 | FGVP-blur (Yang et al., 2023b) | Blur background | 72.3 | 55.8 | 46.5 | 81.3 | 38.6 |
| 14 | SoM (Yang et al., 2023a) | Grounded segments | 61.5 | 47.8 | 45.1 | 75.8 | 57.4 |
| 15 | API (Yu et al., 2024) | Alpha channel fade | 81.6 | 61.1 | *47.4* | 85.8 | 68.4 |
| 16 | ViCrop (Zhang et al., 2025a) | Add object crop | *83.8* | 60.6 | 47.1 | *86.7* | *82.5* |
| 17 | AttWarp *(ours)* | Rectilinear warping | 84.7 | 64.0 | 50.4 | 87.4 | 84.1 |
| 18 | AttWarp-Distill *(ours)* | Efficient inference | 84.1 | 63.1 | 48.9 | 87.2 | 81.8 |
| 19 | AttWarp-Chain *(ours)* | Adaptive Chains | **85.9** | **64.8** | **51.0** | **88.0** | **85.3** |
| 20 | Δ Accuracy | | +4.9 | +2.4 | +3.7 | +1.9 | +8.0 |

- `APIPrompting` (Yu et al., 2024) (Attention-Modulated Image Representation): computes an attention heatmap using an auxiliary VLM (LLaVA or CLIP), and overlays it onto the input image.
- `ViCrop` (Zhang et al., 2025a) (Context Reduction through Attention-Guided Cropping): crops images around regions of high saliency, based on the model's attention map. Notably, `ViCrop` takes two image input *i.e.* both the original and the copped image.

## 4.2 QUANTITATIVE RESULTS

In Tab. 1, we present results for three methods introduced in this work, *i.e.*, `AttWarp`, `AttWarp-Chain`, and `AttWarp-Distill`, evaluated on two MLLMs and five diverse benchmarks.

`AttWarp` **outperforms all baselines on five benchmarks (Tab. 1).** `AttWarp` achieve state-of-the-art results on tasks including text recognition and understanding (TextVQA: LLaVA +8.8%, Qwen +3.7%; DocVQA: LLaVA +7.4%, Qwen +6.8%), compositional and spatial reasoning (GQA: LLaVA +3.2%, Qwen +1.6%), general multimodal question answering (MMMU: LLaVA +3.5%, Qwen +3.1%), and fine-grained understanding and hallucination reduction (POPE: LLaVA +2.2%, Qwen +1.3%). These consistent improvement across diverse tasks arises from `AttWarp`'s capability to highlight task-relevant objects while preserving *global context* and spatial relationships, thus delivering strong

*Question: On the right desk, what is to the left of the laptop?*

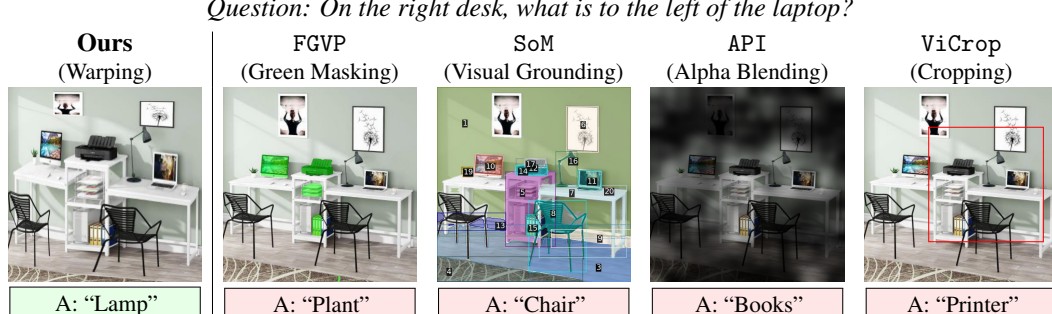

**Figure 4:** `AttWarp` and prior works of image manipulation on the running example. While plausible, prior works are unable to answer the question correctly.

performance on global queries in GQA and fine-grained queries in POPE. Moreover, these results illustrate that `AttWarp`'s *is agnostic to image type* – effective across natural scenes (GQA, TextVQA, POPE), documents (DocVQA), and dense diagrams (MMMU). We provide per-category performance in App. C.1 and study on the extent of warping in App. C.3. Overall, the superior performance of `AttWarp` underscores the significance of enhancing perception for question answering.

`AttWarp` **is plug-and-play with standard MLLMs (Tab. 1).** By default, we evaluate `AttWarp` using LLaVA, consistent with prior works (Zhang et al., 2025a; Yu et al., 2024; Yang et al., 2023a). We further demonstrate `AttWarp` versatility on a stronger MLLM, Qwen2.5-VL. Qwen2.5-VL uses a distinct vision-language fusion strategy with a dedicated cross-modal attention module to integrate visual and textual features, contrasting LLaVA's direct projection of visual features into the language model's input space. The consistent performance gains of `AttWarp` across Qwen2.5-VL and LLaVA-v1.5-7b highlight `AttWarp`'s robust, plug-and-play compatibility with diverse MLLM architectures. Additionally, `AttWarp` achieves similar gains with two other architectures: a dynamic-resolution and pixel-unshuffling approach (InternVL3), and an instruction-aware Q-Former for vision–language alignment (InstructBLIP). See App. D.1 for results.

**Results visual-centric benchmarks:** We further evaluate `AttWarp` on four visual-centric datasets (RealWorldQA, BLINK, MMVP, MIA-Bench). Visual-centric benchmarks are typically harder than text-centric benchmarks because the model must handle challenging pixel inputs (occlusion, small details, counting/localization) and align language with visible evidence, rather than relying mainly on linguistic patterns or memorized text cues. Across all four, `AttWarp` consistently outperforms the base MLLM (Tab. 2), indicating stronger fine-grained visual understand-

**Table 2:** `AttWarp` performance (in %) on visual-centric benchmarks. We report individual accuracy for MMVP, as it aligns with the format of other evaluations. We use LLaVA as the base MLLM. For BLINK, we took the single-image subset.

| LLaVA | MMVP | BLINK | RealWorldQA | MIA |
|---|---|---|---|---|
| Base MLLM | 48.3 | 38.3 | 49.3 | 65.9 |
| AttWarp-Distilled (ours) | 49.3 | 39.7 | 51.1 | 66.4 |
| AttWarp (ours) | 50.7 | 40.4 | 52.1 | 67.8 |
| **AttWarp-Chains (ours)** | **51.0** | **41.2** | **53.1** | **68.8** |
| Δ Accuracy | +2.7 | +2.9 | +3.8 | +2.9 |

ing, more accurate spatial reasoning, and improved visually grounded instruction following. `AttWarp-Distill`, retains most of these gains with lower computational cost, while `AttWarp-Chain`, leverages warping at multiple layers for further improvements. As summarized in Tab. 2, all three variants, `AttWarp`, `AttWarp-Distill`, and `AttWarp-Chain`, improve performance across these four benchmarks, supporting rectilinear attention-guided warping as an effective and general mechanism for enhancing fine-grained visual grounding in MLLMs.

**Chaining provides consistent performance gains over standard `AttWarp` (Tab. 1, rows 7 & 9, rows 17 & 19).** The multi-step `AttWarp-Chain` further boosts performance by iteratively refining attention maps. Examining LLaVA results (**rows 7 & 9**), we observe consistent improvements over `AttWarp` across all five benchmarks (TextVQA +2.2%, GQA +0.7%, MMMU +1.2%, POPE +0.7 %, and DocVQA +2.1%). A similar trend is evident when evaluated on the stronger base model Qwen (**rows 17 & 19**). Qualitatively, Fig. 5 demonstrates how `AttWarp-Chain` adaptively adjusts the warping extent, resulting in enhanced visual grounding and improved task performance.

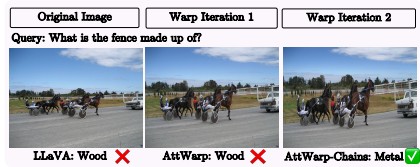

**Figure 5:** `AttWarp-Chain` improves on `AttWarp`

`AttWarp-Distill` **balances accuracy and speed (Tab. 1, Tab. 3).** `AttWarp-Distill` consistently outperforms the base MLLM and matches or exceeds the performance of `ViCrop` (Tab. 3). Optimized specifically for inference efficiency, `AttWarp-Distill` requires only a single MLLM forward pass, making it approximately 3× faster and 2.8× more computationally efficient than `ViCrop`; aligning closely with the computational cost of the base MLLM (8.7 *vs*. 8.5 TFLOPs). In Tab. 3, we compare our method against ViCrop as it achieves comparable performance to `AttWarp-Distill`.

**Table 3:** Comparison of computational overhead. Base MLLM used is LLaVA. Metrics are TFLOPs, peak VRAM (in GB), and number of MLLM passes. Values in brackets show relative cost compared to ViCrop.

| | TFLOPs ↓ | Peak VRAM ↓ | MLLM passes ↓ |
|---|---|---|---|
| ViCrop | 24.2 | 22 | 3 |
| AttWarp-Distill | **8.7 (0.4×)** | **15** | **1** |
| Base MLLM | 8.5 | 15 | 1 |

Other baselines, such as FGVP, SoM, and

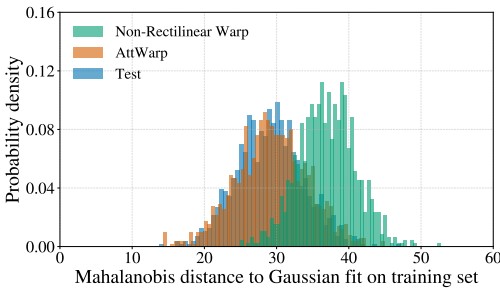

**(a)** Analyzing data shift.

**(b)** Distribution metrics (↓ = lower is better).

| Distribution $\nu$ under: | KID $(\mu_{\text{train}}, \nu) \downarrow$ | FID $(\mu_{\text{train}}, \nu) \downarrow$ |
|---|---|---|
| `AttWarp` | **31.5** | **49.8** |
| Non-Rectilinear Warp | 174.9 | 73.9 |
| Test Set | 19.3 | 56.6 |

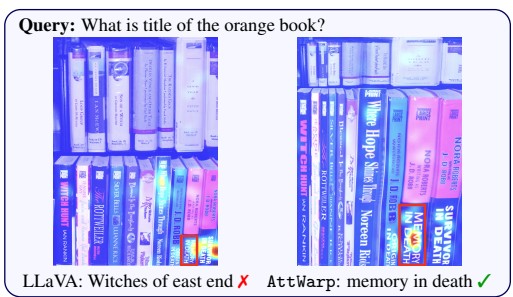

**(c)** Attention alignment with Ground truth boxes.

**(d)** Attention–GT alignment metrics on TextVQA.

| Metric | No warp | With `AttWarp` |
|---|---|---|
| Pointing Game Accuracy ↑ | 37.4% | **42.4%** |
| Proportion ↑ | 0.117 | **0.155** |

**Figure 6:** (a) Mahalanobis distance to the train Gaussian. (b) Train→Test, `AttWarp`, Non-Rect. FID/KID summary. (c–d) Attention–redistribution GT alignment. See Appx. I for setup and additional plots.

APIPrompting, not only perform substantially worse (see Tab. 1), but also incur additional overhead due to multiple inference steps. For more details and cost analysis of `AttWarp` refer App. G.4.

**`AttWarp-Distill` is generalizable (Tab. 1).** We train We train `AttWarp-Distill` on the standard training splits of widely used open-source datasets (TextVQA, GQA, and DocVQA), which form part of the training corpora of most base MLLMs. The substantial improvements on TextVQA (LLaVA +7.9%, Qwen +3.1%), GQA (LLaVA +2.2%, Qwen +0.7%), and DocVQA (LLaVA +4.3%, Qwen +4.5%), demonstrate its strong in-domain generalization capability, whereas its robust performance on POPE (LLaVA +2.1%, Qwen +1.1%), and MMMU (LLaVA +1.9%, Qwen +1.6%), highlights its effective out-of-domain generalization. Further details on the training procedure and marginal prediction in App. G.3.

### 4.3 ABLATIONS AND ANALYSIS

**Warping Improves MLLMs Attention.** To gauge how `AttWarp` reshapes the *spatial faithfulness* of the internal attention of the model, we adopt two widely-adopted localization metrics that rely only on ground-truth boundary boxes (bbox) and the heat map itself: 1) **Pointing Game Accuracy (top-1 attention in bbox)**: checks whether the *single* most salient pixel of the attention map falls within the GT bbox (Zhang et al., 2016). 2) **Proportion (fraction attention within box)**: the fraction of *total* attention mass that lands inside the bbox (Wang et al., 2020). As observed in Tab. 6d, the $+5\%$ relative jump in *Pointing Game Accuracy* and the $+3.8\%$ boost in *Proportion* on TextVQA confirm that our rectilinear warp *tightens* the focus of the model: the attention mass and dominant peak shifts to the task relevant region. These findings indicate that `AttWarp` improves the MLLM performance because of better attention distribution (see Fig. F and App.F). We further extend this study and empirically verify that `AttWarp` *expands the correct image regions*. We quantify this by checking how often `AttWarp` expands the bboxes of the relevant regions (refer App. E for more details).

**`AttWarp`'s rectilinear design preserves the image distribution.** Pixel–level transforms risk shifting test images away from the manifold on which the model was trained. We probe this risk for LLaVA by first extracting ViT-L/14 CLIP features (Radford et al., 2021) for $12k$ randomly sampled GQA-train images and fitting a full-covariance Gaussian to those embeddings. Figure 6a compares the resulting *Mahalanobis distance* histogram for three image sets: (i) Test images (blue); (ii) images warped by `AttWarp` (orange); and (iii) a non-rectilinear warping baseline, inspired by (Recasens et al., 2018) (green). `AttWarp` almost exactly overlaps the unmodified test distribution (both peak at $\approx 29\sigma$), whereas distribution based on Non-Rectilinear Warp shifts rightward (peak $\approx 37\sigma$) and exhibits a heavier tail, indicating a measurable distribution drift. We also test this aspect using standard two-sample metrics, *FID* (Heusel et al., 2017) and *KID* (Bińkowski et al., 2018) between the training

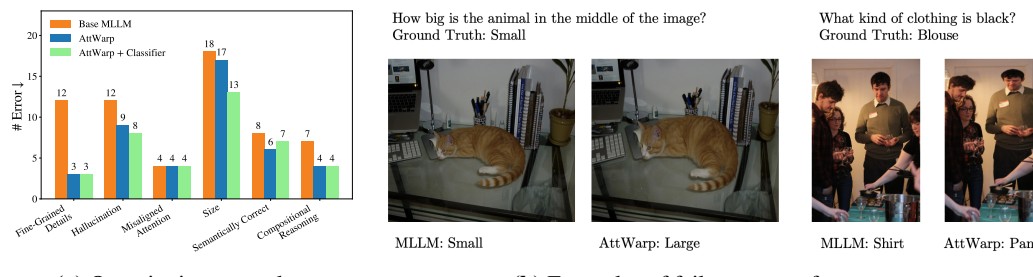

(a) Quantitative error plot.

(b) Examples of failure cases of `AttWarp`.

**Figure 7: Error Characterization and Failure Cases.**

distribution and each test variant. As evident from Table 6b, `AttWarp` achieves significantly better alignment with the training distribution (**31.5** vs. 174.9 KID and **49.8** vs. 73.9 FID), effectively reproducing the train–test baseline gap. Alongside this, as can be seen in Figure 6a, our approach closely matches the baseline metrics indicating that `AttWarp` preserves the underlying image manifold and introduces negligible distribution shift (see App. I for graphs and details).

**Error Analysis.** We randomly sampled 150 VQA tasks from the GQA and TextVQA evaluation. A total of 61 were incorrect for the base LLaVA model, and 42 were incorrect for `AttWarp`. We bin these into the closest failure modes: *Fine-Grained Details*, where the answer is very small in size; *Hallucination* when the answer includes details not present in the image; *Misaligned Attention*, when focus shifts to the wrong object; *Size*, for questions involving object scale; *Semantically Correct*, when the answer is correct but phrased differently; and *Compositional Reasoning*, involving multiple objects and relationships. Fig. 7a shows fewer errors in fine-grained and compositional cases. However, we note that warping can sometimes suppress peripheral context needed for global reasoning, and performance may degrade if the underlying attention is noisy. While absolute object sizes are changed, relative proportions are preserved, limiting errors in size-related tasks.

**Reducing the Error Modes.** The error analysis shows that AttWarp reduces errors across all categories compared to the base MLLM (e.g., hallucination: $12 \rightarrow 9$, fine-grained details: $12 \rightarrow 3$). However, AttWarp remains susceptible to errors related to object size and misaligned attention, motivating the development of methods that further enhance robustness. To explicitly target these failure cases, we design a simple classifier that decides whether to apply AttWarp. Concretely, we reuse AttWarp-Distill's weights and network, and replace its last two layers with a binary classifier head. We create a training set by evaluating AttWarp on the training split of AttWarp-Distill (App. G), and use these outcomes to train the classifier. We denote the resulting classifier-gated variant by AttWarp[†], i.e., AttWarp[†] = Classifier + AttWarp. We then perform the same error analysis for AttWarp[†], shown in Fig. 7a (green bars). AttWarp[†] further reduces errors in size and hallucination while keeping other categories essentially unchanged. Misaligned-attention errors remain unchanged, as both the base MLLM and AttWarp fail on the same underlying attention misalignment. Designing mechanisms to correct attention misalignment is, therefore, a productive direction for future work.

## 5 CONCLUSION

We introduced `AttWarp`, a plug-and-play, test-time self-correction mechanism that uses an MLLM's cross-modal attention to rectilinearly resample the input, expanding query-relevant regions while preserving global context. Without changing model weights or architecture, it consistently improves accuracy, spatial grounding, and hallucination rates across nine benchmarks and four MLLMs. By intervening before feature extraction, `AttWarp` complements internal attention refinements and shows that input-level, information-preserving transformations can help the same models see better.

## 6 ACKNOWLEDGEMENT

This research is based upon work supported by U.S. DARPA ECOLE Program No. HR00112390060. The views and conclusions contained herein are those of the authors and should not be interpreted as necessarily representing the official policies, either expressed or implied, of DARPA or the U.S. Government. The U.S. Government is authorized to reproduce and distribute reprints for governmental purposes, notwithstanding any copyright annotation therein. This work has also received support from the School of ICS at UC Irvine and from an NSF grant CCF 2348624.

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

APPENDIX

This appendix provides additional material to support and extend the findings in the main paper. Each entry is clickable.

# A FAQ

1. How is distributional integrity maintained during warping?

   **Ans.** Two primary reasons underlie this preservation.
   First, the rectilinear warping preserves spatial attributes, positional relationships, and structural integrity. At the tokenization stage, objects maintain their semantic identities. Second, the rectilinear warping directly aligns with the data augmentation strategy used during CLIP pre-training. Specifically, CLIP pre-training uses RandomResized-Crop—randomly sampling 8–100% of the image with aspect ratios 3/4–4/3, then resizing to 224×224 px—scaling rows and columns independently yet axis-aligned, thus exposing the model to varied scales and positions for robust, scale-invariant representations Radford et al. (2021). As the warp is defined as

   $$(x', y') = (F_x(x), F_y(y)),$$

   with monotone 1-D CDFs, its Jacobian is strictly diagonal. Consequently, (1) each ViT patch is subjected only to axis-wise scaling ($F_x'(x)$ along $x$, $F_y'(y)$ along $y$), and (2) borders remain orthogonal. Hence, what the encoder encounters at the token level in `AttWarp` is identical in form to the resize-then-crop augmentation used during CLIP training, albeit with locally different scale factors. Thus, rectilinear warping keeps token statistics on the same manifold learned during pre-training.

2. Did you use a single layer or multiple layers? The formulation you presented appears to be for a multi-layer approach.

   **Ans.** While the formulation demonstrates a general multi-layer capability, our quantitative and qualitative analyses indicated that a single layer (specifically the 16th layer for the Qwen and the 20th layer for the LLaVA model) performed better than an average across all layers in this instance.

3. If the attention is highly inaccurate, then what?

   **Ans.** If the attention map is highly inaccurate, both the base MLLM and the MLLM with AttWarp tend to fail and produce incorrect answers. However, in most practical scenarios, the attention maps are only moderately suboptimal. In these cases, AttWarp is particularly effective—its warping mechanism enhances the attention distribution, leading to improved performance. Detailed experimentation focused on robustness are presented in Appendix B.5.

4. What happens when there are multiple objects of focus? Does `AttWarp` work in that case?

   **Ans.** As demonstrated in Appendix E.1, we conducted a study on cases with multiple objects of focus—ranging from two up to five distinct regions. Our results show that `AttWarp` consistently expands the target regions, with 89% of the annotated bounding boxes exhibiting increased area after transformation. In many instances, all objects of interest (up to five per example) are effectively expanded, highlighting the method's robustness in handling complex, multi-object referential expressions.

5. What if image warping largely distorts the image?

   **Ans.** The extent of distortion introduced by image warping is governed by the choice of transformation function $\mathcal{T}$. For the transformations we utilize—$\mathcal{T} \in \{\mathrm{sqrt}, \mathrm{identity}, \mathrm{square}\}$—significant distortion is not observed in practice. Extreme warping only occurs if the attention map is highly concentrated on a single, small region, which is rare and typically corresponds to extremely fine-grained queries. In such cases, the resulting distortion is often advantageous, as it further magnifies the most relevant region, thereby improving the model's ability to answer the query.

# B    ABLATION STUDY ON ATTENTION SCORE MATRIX

## B.1    LAYER SELECTION ACROSS MLLM LAYERS

To select the attention map that best captures the visual semantic signal of fine-grained image details with respect to the query, we compared two normalization schemes—*absolute* and *relative* attention—through every cross attention layer of *LLaVA-1.5-7B*. The absolute map is the raw cross-attention weight assigned to each image token when the model answers the question, whereas the relative map divides this weight by a *caption-only* baseline obtained from an auxiliary forward pass that asks the model to "describe the image briefly." The intuition is that relative normalization suppresses static scene priors (e.g. sky or grass that invariably attract some attention) and instead highlights query-specific regions. We benchmark both variants on 2000 TextVQA validation images using **Pointing-Game Top-1 Hit** (Zhang et al., 2016) and **AM@all** (Wang et al., 2020). Fig. 8 presents a *layer-wise* localization analysis for LLaVa 7b: scores increase through the mid-network for both schemes, peak with the *absolute* map from layer 20 (TOP-1 $\approx 0.36$, AM@ALL $\approx 0.22$), and then plateau or decline; relative attention helps in earlier layers but never exceeds the absolute variant beyond layer 18 and likewise weakens past layer 20. These findings indicate that the deeper blocks progressively concentrate the model's "attention budget" on the object of interest, but the final layers begin to divert attention to token generation. Consequently, all LLaVA-based warps in our work `AttWarp` employ the absolute attention of layer 20. An analogous sweep on *Qwen-VL* reveals its optimum at layer 16. The qualitative evidence in Fig. 9 mirrors the quantitative trend: attention tightens around the "yamaha" logo on the bass drum from layers 10 to 20, underscoring the utility of this layer-wise study - previously unexplored in prior works and guiding our choice of layer 20 for LLaVA-based models and layer 16 for Qwen-VL as the most reliable attention sources.

**Practical Considerations against Adaptive Layer Selection -** Dynamically evaluating multiple layers per query would add significant runtime overhead, undermining the low-latency goals of test-time adaptation. Our focus in this study was to validate the core AttWarp mechanism under the best static choices.

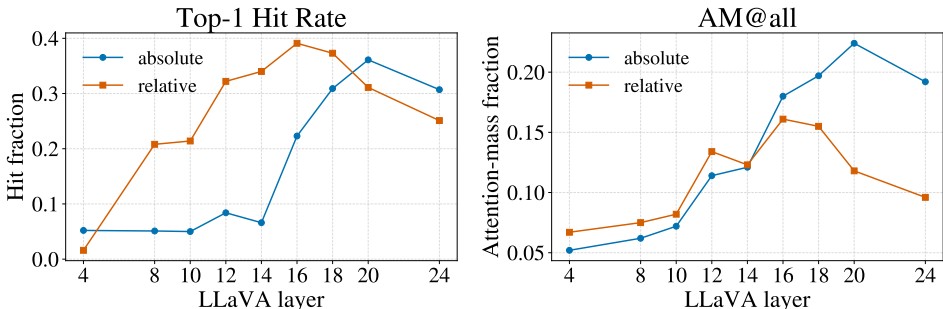

**Figure 8:** Layer-wise localisation quality of LLaVA-1.5-7B cross-attention maps on TextVQA Zhang et al. (2025a) images. Curves report **Top-1 Hit Rate** (Zhang et al., 2016) (left) and **AM@all** (Wang et al., 2020) (right) for absolute (blue) and relative (orange) attention.

*Question:* What is the brand of the bass drum? *Answer:* Yamaha

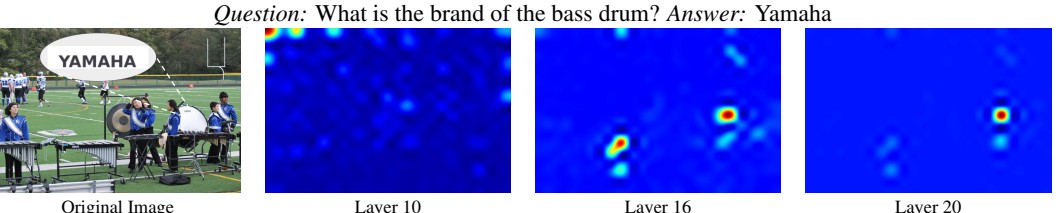

**Figure 9:** Below (left to right): (a) Original image, zoomed in at yamaha (answer), (b) to (d) attention maps captured from layers 10, 16, 20 respectively. As can be seen, the attention localization improves drastically from layer 10 to 20, indicating improved query-specific spatial understanding.

| LLaVA | | | | | |
|---|---|---|---|---|---|
| Method | TextVQA | GQA | MMMU | POPE | DocVQA |
| Base MLLM | 49.3 | 60.5 | 36.9 | 85.3 | 18.1 |
| AttWarp (Layer 20) | 58.1 | 63.7 | 40.4 | 87.5 | 25.5 |
| AttWarp (Layer 22) | 58.4 | 62.8 | 39.1 | 87.1 | 24.9 |
| Qwen | | | | | |
| Method | TextVQA | GQA | MMMU | POPE | DocVQA |
| Base MLLM | 81.0 | 62.4 | 47.3 | 86.1 | 77.3 |
| AttWarp (Layer 16) | 84.7 | 64.0 | 50.4 | 87.4 | 84.1 |
| AttWarp (Layer 22) | 84.2 | 63.6 | 49.8 | 87.6 | 83.9 |

**Table 4:** Performance of AttWarp across different attention layers on LLaVA and Qwen backbones.

**Robustness to Attention Layer Choice (Table 4)**: These results reveal a simple and robust rule for selecting attention maps in MLLMs. As previously observed in CNNs and ViTs, MLLMs too exhibit a similar trait: deeper layers produce attention maps that become increasingly centered on objects of interest. Specifically, we find that attention maps from layers with depth $\geq 15$ in MLLMs consistently yield task-appropriate, region-of-interest-aligned maps for AttWarp. To validate robustness to the specific choice of layer, we re-run AttWarp using a new, much deeper layer (24) for both LLaVA and Qwen and evaluate on five benchmarks. Across all datasets, the layer-24 variants closely match or slightly exceed the gains obtained with layers 16/20, demonstrating that AttWarp is largely insensitive to the precise layer index as long as it is sufficiently deep. In practice, we therefore recommend using an earlier deep layer (e.g., 16–20) to reduce computational overhead while preserving the full performance benefits.

### B.2    ATTENTION AGGREGATION STUDY

**Robustness of Attention Head Aggregation**    Each attention head captures a different aspect of the multimodal interaction (e.g., color, shape, texture, positional cues). Beyond selecting the optimal layer, we analyzed the strategy for aggregating attention across heads. Our analysis confirms that uniformly averaging attention from all heads provides the most robust and informative signal for warping. As shown in Table 5, this approach significantly outperforms alternatives like max-pooling or using random subsets of heads, validating our data-driven design choice.

**Table 5:** Effect of attention head aggregation strategy on TextVQA accuracy (%). Our method of averaging all heads is the most effective and robust.

| Aggregation Methodology | TextVQA Accuracy |
|---|---|
| Mean over all 32 heads (Ours) | **58.1** |
| Max-pooling across 32 heads (token-wise) | 55.3 |
| Mean over 8 randomly selected heads | 54.6 |
| Random single head (re-sampled per run) | 51.9 |

### B.3    PIXEL-SPACE VS. FEATURE-SPACE WARPING

An alternative to our proposed image warping is to directly manipulate the internal feature space of the multimodal LLM. We explored this option by injecting a bias into the hidden states after normalization in the first cross-attention block:

$$\mathbf{h}' = \text{LayerNorm}(\mathbf{h}) + \lambda \cdot \mathbf{b},$$

where $\mathbf{b}$ is constructed from attention weights and $\lambda$ controls the bias strength. Although conceptually attractive, this approach proved unstable in practice: performance gains were marginal (only **+0.6%** on TextVQA). Moreover, the results of the feature space warping were highly sensitive to the choice of $\lambda$. As highlighted in previous research Sun et al. (2025), direct manipulations at the internal feature level inherently risk causing significant distribution shifts that interact unpredictably with architectural components such as pre-or post-normalization layers.

In contrast, our rectilinear warping *pixel-space* avoids these pitfalls and offers several concrete benefits.

- **Stability and robustness:** The internal computation of the model remains unchanged, ensuring consistent behavior across datasets and queries.

- **Interpretability:** The warp is visually transparent: the expanded regions correspond directly to areas of high attention, allowing intuitive inspection and debugging.

- **Architecture agnostic:** No architectural modifications are needed, making the approach compatible across diverse MLLMs and even with external attention sources.

- **Geometry preservation:** The axis-wise CDF warp expands relevant areas while maintaining global structure and relative spatial layout, safeguarding spatial reasoning ability.

## B.4 Effect of the Transform Hyperparameter ($\mathcal{T}$) in AttWarp

To assess the role of the transform function $\mathcal{T}$, we performed an ablation across several functional forms. Across both TextVQA and GQA (Tables 6 and 7), results are stable for simple choices such as square root, identity, square, and cubic. The identity and square transforms consistently achieve the best accuracy (58.1–58.3 on TextVQA; 63.3–63.5 on GQA), while alternatives yield only slightly lower scores. These findings show that AttWarp is robust to the choice of $\mathcal{T}$ and that the identity transform serves as a strong default—delivering state-of-the-art accuracy without dataset-specific tuning and outperforming competitive baselines such as ViCrop.

**Table 6:** Effect of the transform function $\mathcal{T}$ on TextVQA accuracy (%).

| Transform | Accuracy (%) |
|---|---|
| LLaVA Baseline | 49.3 |
| $\mathcal{T}$ = sqrt | 56.8 |
| $\mathcal{T}$ = Identity | **58.1** |
| $\mathcal{T}$ = square | **58.3** |
| $\mathcal{T}$ = cube | 57.8 |

**Table 7:** Effect of the transform function $\mathcal{T}$ on GQA accuracy (%).

| Method | LLaVA | Qwen-VL |
|---|---|---|
| Baseline Model | 60.5 | 62.4 |
| ViCrop | 60.9 | 60.6 |
| AttWarp (Identity) | **63.3** | **63.5** |
| AttWarp (sqrt) | 63.7 | 64.0 |

## B.5 Impact of Attention Bias and Robustness to Corruptions and Adversarial Perturbations

A natural concern is that the performance of AttWarp depends on the underlying pretrained model. If the attention is biased—for example, by over-focusing on high-frequency features—this may limit the generalization of our method. Similar to other test-time adaptation (TTA) approaches for multimodal LLMs, such as ViCrop, our method leverages attention maps to highlight query-relevant regions and is therefore subject to the same dependency. It is important to emphasize, however, that AttWarp is designed as a TTA mechanism to enhance accuracy for visual question answering (VQA), rather than a debiasing approach to correct distribution shift in the model parameters.

To examine robustness under conditions where attention may be less reliable, we conducted additional experiments using synthetic corruptions following the standard IMAGENET-C protocol Hendrycks & Dietterich (2019). Specifically, we evaluated impulse noise, Gaussian noise, and shot noise to depict aggressive high-frequency injections applied to the TextVQA dataset. All corrupted images were resized to the standard $512 \times 512$ resolution for evaluation, and no retraining or adaptation beyond test-time warping was performed.

**Table 8:** Accuracy (%) of LLaVA and LLaVA+AttWarp under different image corruptions, following the IMAGENET-C protocol.

| Corruption | LLaVA | LLaVA + AttWarp |
|---|---|---|
| Impulse noise | 36.8 | **40.4** |
| Gaussian noise | 37.6 | **41.0** |
| Shot noise | 36.0 | **39.8** |

These results demonstrate that while AttWarp inherits the biases of the attention maps it uses, it still provides consistent accuracy gains even under corruption. This analysis shows that AttWarp is not a

debiasing method but remains a reliable and effective TTA mechanism to improve VQA performance in both clean and noisy settings.

**Robustness to Adversarial Perturbations.** To further probe resilience, we introduce targeted adversarial perturbations specifically engineered to misdirect the MLLM's initial attention distribution away from semantically pertinent regions (Fig. 10, top). This adversarial setting provides a stringent test of `AttWarp` 's iterative refinement capability (Sec. 3.2). As illustrated in Fig. 10 (bottom), while the initial adversarial attack successfully perturbs attention and induces a faulty warp, subsequent iterations progressively correct the transformation, re-aligning focus with the relevant visual content. This behavior highlights a critical self-correction mechanism: iterative `AttWarp` is able to recover from compromised attention signals, ensuring more reliable grounding even under adversarial conditions.

*Question:* What brand liquor is on the right? *Answer:* Bowmore Islay

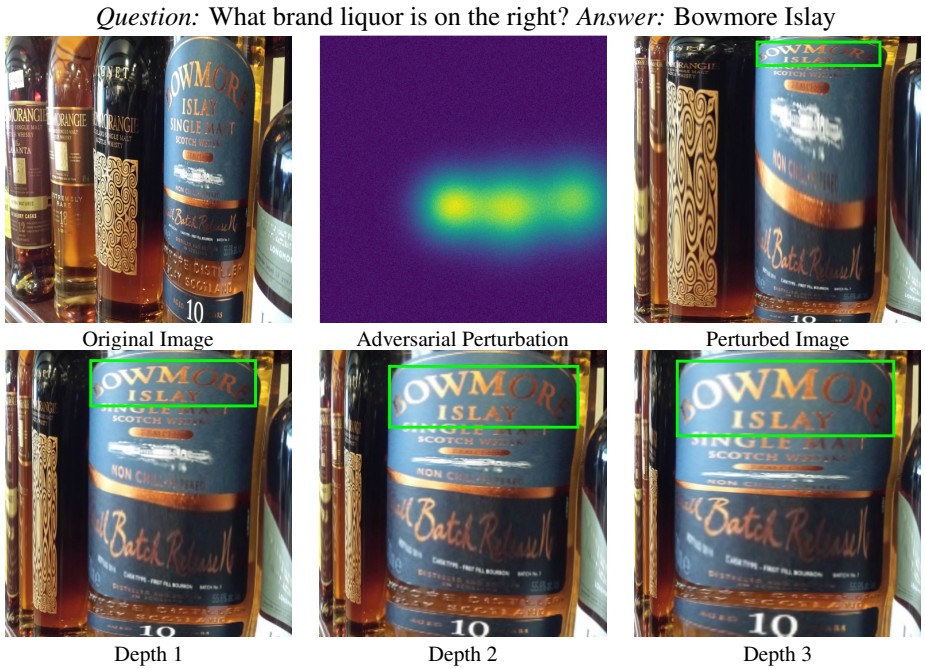

**Figure 10:** (top row) the original image, the adversarial perturbation designed to corrupt attention, and the resulting perturbed image in which the task-relevant region is shrunk; (bottom row) outputs of `AttWarp` at Depth 1–3, illustrating how our method progressively overcomes the interference to refocus on the relevant region.

# C    TASK SPECIFIC ANALYSIS OF `AttWarp`

## C.1    TASK CATEGORIES, BENEFITS, AND LIMITATIONS

`AttWarp` magnifies the query-relevant region while preserving relative object positions through its rectilinear design. This leads to clear gains in three categories: **fine-grained perception**, **spatial reasoning**, and **hallucination mitigation**. Table 9 reports the improvements across datasets. Across GQA's 15 semantic sub-categories, `AttWarp` improves performance consistently. The only exception is a marginal drop of **0.2%** in the *compare-attribute* sub-category involving direct size comparisons. This remains the sole degradation observed, underscoring the reliability of the approach.

**Table 9:** Category-level improvements of `AttWarp`. Accuracy in %. Best results in **bold**.

| Task Category | Dataset | LLaVA | AttWarp | Δ |
|---|---|---|---|---|
| Fine-grained perception | TextVQA (fine-grained) | 49.9 | **59.6** | +9.3 |
| | DocVQA (table/list/form/handwritten) | 13.6 | **19.5** | +5.9 |
| Spatial reasoning | TextVQA (spatial) | 54.5 | **64.9** | +10.4 |
| | DocVQA (layout) | 29.4 | **37.6** | +8.2 |
| | MMMU (spatial) | 38.2 | **44.8** | +6.6 |
| | GQA (relation) | 51.5 | **56.4** | +4.9 |
| Hallucination mitigation | POPE | 85.3 | **87.5** | +2.2 |
| | GQA (object) | 86.1 | **89.4** | +3.3 |

## C.2 DISTORTION AND UPPER-BOUND CONCERNS

A concern is whether warping distorts shapes or positional reasoning. By design, `AttWarp` preserves grid alignment and relative geometry. This is reflected in GQA's semantic dimensions shown in Table 10, where accuracy improves across all categories.

**Table 10:** Performance on GQA semantic categories. Accuracy in %.

| Category | LLaVA | AttWarp |
|---|---|---|
| Relation | 51.5 | **56.4** (+4.9) |
| Attribute | 67.8 | **69.3** (+1.5) |
| Category | 51.7 | **55.1** (+3.4) |
| Object | 86.1 | **89.4** (+3.3) |
| Global | 62.5 | **65.5** (+3.0) |

Manual sub-sampling from other datasets confirms similar gains. AttWarp boosts positional reasoning and object shape sensitivity across TextVQA, DocVQA, and MMMU as shown in Table 11.

**Table 11:** Cross-dataset improvements on positional reasoning and object shape. Accuracy in %.

| Dataset | Category | LLaVA | AttWarp | Δ |
|---|---|---|---|---|
| TextVQA | Positional reasoning | 54.5 | **64.9** | +10.4 |
| DocVQA | Layout (positional) | 29.4 | **37.6** | +8.2 |
| MMMU | Object shape | 36.6 | **40.7** | +4.1 |
| DocVQA | Diagram (object shape) | 18.8 | **22.2** | +3.4 |

These results show that AttWarp enhances rather than constrains reasoning involving spatial relations, positions, and shapes.

## C.3 GLOBAL CONTEXT AND WARPING INTENSITY

For questions requiring global context, AttWarp adapts the strength of warping to the input. We quantify this with a warping intensity metric, defined as the mean log-change of grid-cell Jacobians. Fine-grained queries undergo stronger magnification, while global queries are only mildly affected.

**Table 12:** Warping intensity by question type. Values are mean log-change of Jacobian determinants.

| Category | $\mathbb{E}[\Delta \log |\det \mathbf{J}|]$ |
|---|---|
| Attribute | 0.19 |
| Category | 0.18 |
| Global | 0.16 |
| Object | 0.19 |
| Relation | 0.22 |
| Fine-grained | **0.26** |

Fine-grained questions produce a $\sim$**30%** area change, amplifying small detail-rich regions. Global questions produce only a $\sim$**17%** change, preserving scene layout while still enabling mild emphasis. This explains performance increase on global tasks.

### C.4 TASK-WISE COMPARISON WITH VICROP

ViCrop uses dual inputs (original and cropped images). A natural concern is whether this dual-image design provides an advantage. Table 13 shows that `AttWarp` consistently outperforms ViCrop across semantic categories on GQA.

**Table 13:** Comparison with ViCrop on GQA semantic categories. Accuracy in %. Best results in **bold**.

| Category | LLaVA | ViCrop | AttWarp |
|---|---|---|---|
| Relation | 51.5 | 51.9 | **56.4** |
| Attribute | 67.8 | 68.2 | **69.3** |
| Category | 51.7 | 52.0 | **55.1** |
| Object | 86.1 | 86.3 | **89.4** |
| Global | 62.5 | 63.4 | **65.5** |

This analysis shows that AttWarp not only avoids the pitfalls of dual-image inputs but also provides consistent improvements across all semantic dimensions.

## D ADDITIONAL EXPERIMENTS

### D.1 EXTENDED GENERALIZATION EXPERIMENTS

We further examined the generalizability of `AttWarp` by applying it to two additional multimodal LLMs: the recently released **InternVL-3 8B** (opensourced in April 2025) Zhu et al. (2025) and **InstructBLIP** Dai et al. (2023). For both models, attention maps were extracted following the same protocol used for LLaVA and Qwen.

**InstructBLIP -** On TEXTVQA and GQA, `AttWarp` improves over both the baseline InstructBLIP and the ViCrop baseline. Moreover, the chained variant (`AttWarp-Chain`) yields additional gains, confirming that iterative refinement can further sharpen attention-driven warping.

**InternVL-3 8B -** On TEXTVQA and GQA, `AttWarp` consistently outperforms both the strong baseline InternVL-3 and ViCrop. These results indicate that the benefits of our approach transfer effectively even to cutting-edge architectures trained at scale.

**Table 14:** Generalization of `AttWarp` across **InstructBLIP** and **InternVL-3**. Evaluation metric: accuracy (%). Best results in **bold**.

| Method | InstructBLIP | | InternVL-3 8B | |
|---|---|---|---|---|
| | **TextVQA** | **GQA** | **TextVQA** | **GQA** |
| Baseline | 35.2 | 49.4 | 80.2 | 61.4 |
| ViCrop | 46.6 | 49.7 | 82.7 | 63.9 |
| `AttWarp` | 47.8 | 51.3 | **84.6** | **65.9** |

Together, these extended evaluations reaffirm the robustness of `AttWarp` across a diverse set of multimodal LLMs—spanning both established architectures (InstructBLIP) and the latest state-of-the-art models (InternVL-3). The consistent improvements over strong baselines and ViCrop highlight `AttWarp` as a broadly applicable, model-agnostic test-time adaptation mechanism.

## D.2 FINE-GRAINED AND CATEGORY-WISE RESULTS ON DOCVQA

To further evaluate the capacity of `AttWarp` for both fine-grained recognition and global structural reasoning, we analyze its performance on the DocVQA dataset across diverse structural categories. These include *Free Text*, *Table*, *Layout*, *Form*, *Handwritten*, and *Diagram*, which together capture the spectrum of challenges in document question answering. All experiments are conducted on images resized to $512 \times 512$, since original DocVQA images are much larger. This resizing ensures computational feasibility across methods. In particular, for dual-image methods such as ViCrop, processing full-resolution images creates prohibitively many tokens, making the resized setting especially relevant for fair comparison.

**Table 15:** Category-wise accuracy (%) on DocVQA using `Qwen2.5-VL-7B` as the base MLLM. All results are reported on images resized to $512 \times 512$ to ensure computational feasibility across methods. `AttWarp` consistently outperforms all baselines across every document structure category except Yes/No, where ViCrop performs best. Gains span both fine-grained (Form, Hand-written, Table) and global (Layout, Diagram) reasoning categories.

| Method | Overall | Free Text | Table | Layout | Form | Hand-written | Diagram | Others | Image | Y/N |
|---|---|---|---|---|---|---|---|---|---|---|
| Qwen | 77.8 | 78.1 | 76.3 | 85.6 | 75.8 | 63.2 | 79.6 | 80.0 | 71.4 | 82.1 |
| API | 68.4 | 68.9 | 60.8 | 79.5 | 68.9 | 60.9 | 72.1 | 80.0 | 66.1 | 64.3 |
| ViCrop | 82.3 | 83.0 | 78.8 | 87.2 | 77.9 | 68.3 | 80.8 | 84.8 | 76.5 | **96.4** |
| `AttWarp` | **84.1** | **86.2** | **79.1** | **89.9** | **83.5** | **71.4** | **81.5** | **86.7** | **82.1** | 92.9 |

Table 15 shows that `AttWarp` delivers consistent and significant improvements over all baselines, not only in overall accuracy but also within every category. The method achieves particularly strong gains in challenging fine-grained settings such as *Form* (+5.6) and *Handwritten* (+8.2), which require precise localization and interpretation. At the same time, improvements in global categories such as *Layout* (+2.7) and *Diagram* (+0.7) highlight its ability to capture holistic structural cues. The only exception is the Yes/No category, where ViCrop slightly outperforms our approach. Importantly, these results were computed in the resized $512 \times 512$ setting, which avoids the prohibitive token explosion of large original images for dual-image methods like ViCrop, thereby ensuring fair and tractable comparisons. Overall, these findings demonstrate the robustness and versatility of `AttWarp` for document question answering tasks spanning both local detail and global structure.

## D.3 STABILITY AND TERMINATION IN ATTWARP-CHAINS

This subsection provides additional results complementing the discussion in Section 3.2, where we introduced an adaptive stopping criterion for iterative warping. Since AttWarp-chains operates recursively, each warped input influences subsequent attention maps. While this iterative refinement initially sharpens focus and improves accuracy, excessive iterations can lead to over-expansion, noise, and performance degradation. This underscores the importance of a principled termination rule.

Figure 11 illustrates this phenomenon on TEXTVQA (blue curve) and GQA (orange curve). Accuracy improves significantly in the first few iterations—rising from ~47% to ~60% on TEXTVQA, and from ~60% to ~64% on GQA—but then plateaus and declines once the depth exceeds two to three iterations. This decline reflects the recursive instability of fixed-length warping. By contrast, the adaptive `AttWarp-Chain` (dashed lines) consistently outperforms all fixed-depth settings by halting precisely when attention distributions stabilize.

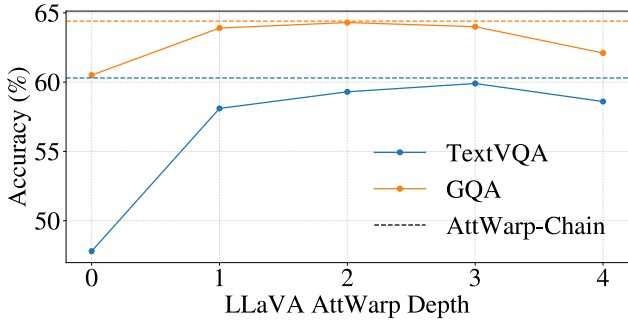

**Figure 11:** Impact of iterative warping depth on accuracy for TEXTVQA and GQA. Fixed-length warping initially improves performance but degrades with excessive iterations due to recursive instability. The adaptive `AttWarp-Chain` (dashed lines), guided by the KL divergence stopping criterion, consistently achieves the best accuracy while avoiding over-warping.

The adaptive stopping rule monitors changes in successive attention maps and terminates the chain once the distributions converge, as formalized by the KL divergence threshold in Eq. 6. This ensures that refinement halts exactly when attention has sufficiently concentrated on query-relevant regions. In practice, `AttWarp-Chain` not only prevents instability but also achieves higher accuracy than any fixed-depth configuration while being computationally more efficient by avoiding unnecessary iterations.

### D.4 EXTERNAL ATTENTION MAPS

`AttWarp` **is compatible with attention from** *external* **models.** All prior experiments applied `AttWarp` using *internal* cross-attention maps, *i.e.* extracted from the same MLLM used for downstream inference. Here we probe another axis of generalization: can model A (e.g., LLaVA) benefit from an image warp constructed from the attention maps of model B? Demonstrating such flexibility would establish `AttWarp` as a model-agnostic, general-purpose mechanism independent of the substrate attention provider.

To investigate this, we source attention maps from two different classes of external models: (i) the text-to-image generative backbone Stable Diffusion 2.1 Rombach et al. (2022), and (ii) a strong multimodal LLM, Qwen-VL Yang et al. (2024a). The warped images are then processed by LLaVA for downstream tasks. Results in Table 16 show that `AttWarp` consistently improves performance across both TEXTVQA and GQA, even when relying on external substrate attention. Notably, MLLMs serve as better sources than generative vision models: attention from Qwen-VL yields the strongest gains (+10% on TEXTVQA, +3.4% on GQA), while Stable Diffusion also provides improvements though of smaller magnitude (+6.7%, +2.2%). Interestingly, even external MLLM attention outperforms the base LLaVA, underscoring the benefit of stronger substrate models. Complementary experiments transferring attention in the opposite direction *i.e.* from a weaker LLaVA-7B to strengthen a larger LLaVA-34B—are reported in Appendix E.3.

**Table 16:** Performance of `AttWarp` using attention from internal vs. external models. Evaluation metric: accuracy (%). Best results in **bold**.

| Method (Source of Attention) | TextVQA | GQA |
|---|---|---|
| Base LLaVA | 49.3 | 60.5 |
| + `AttWarp` (Internal: LLaVA) | 58.1 | 63.7 |
| + `AttWarp` (Stable Diffusion Rombach et al. (2022)) | 56.0 | 62.7 |
| + `AttWarp` (Qwen-VL Yang et al. (2024a)) | **59.3** | **63.9** |

These findings demonstrate that `AttWarp` is not restricted to internal attention but generalizes robustly across external sources as well, with stronger multimodal LLMs providing the most effective substrate attention maps.

# E    BEYOND STANDARD VQA: EXTENDED EXPERIMENTS

In this section, we present several forward-looking experiments to probe the broader applicability and potential of `AttWarp`. We demonstrate its effectiveness in expanding all query-relevant regions, generalizing to open-vocabulary object detection, and even improving the performance of larger models using attention maps from smaller ones. Collectively, these results highlight the versatility and extensibility of our approach beyond standard VQA settings.

## E.1    ATTWARP EXPANDS ALL QUERY-RELEVANT REGIONS

We evaluate whether `AttWarp` effectively enlarges query-relevant regions using ground-truth bounding boxes from prior datasets. Results show that our method consistently expands the salient regions, both for single- and multi-object queries.

**Table 17:** Expansion of query-relevant bounding boxes under `AttWarp`.

| Dataset | % Boxes Expanded | Mean Area Increase |
|---|---|---|
| TextVQA (single-region) | **94.0** | **+76%** |
| gRef (multi-region) | **88.6** | **+39%** |

For TextVQA (Zhang et al., 2025a), `AttWarp` expanded **94%** of salient bounding boxes, with an average area increase of **76%**, confirming its ability to magnify relevant image regions.

To test cases with multiple objects of focus, we used the gRef dataset (Liu et al., 2023a), which provides multiple ground-truth bounding boxes per query. Here, `AttWarp` expanded **88.6%** of target boxes with a mean area increase ratio of **1.39** (**+39%** area).

These results demonstrate that `AttWarp` reliably enlarges query-relevant regions, including complex multi-object settings, while preserving their spatial grounding.

## E.2    OPEN-VOCABULARY OBJECT DETECTION

We next evaluate the versatility of `AttWarp` on Open-Vocabulary Object Detection (OVOD), a setting that extends beyond VQA. In OVOD, the goal is to localize objects in images based on free-form referring expressions rather than a fixed label set. We use a dataset of 10,000 examples to test whether our approach improves localization under this challenging setup.

Our pipeline applies `AttWarp` with LLaVA-7B, guided by each referring expression. The warped image is then processed by the LISA-LLaVA-7B framework Lai et al. (2024) to predict bounding boxes, which are mapped back to the original image space using the inverse warp. This enables direct IoU-based evaluation against ground-truth annotations.

**Table 18:** OVOD performance with LISA-LLaVA-7B. Accuracy in %.

| Model | Original Images | Warped Images | $\Delta$ |
|---|---|---|---|
| LISA-LLaVA-7B | 54.0 | **61.0** | +7.0 |

Representative qualitative examples are shown in Fig. 12. These results confirm that `AttWarp` extends effectively to OVOD, improving localization accuracy by **+7%**.

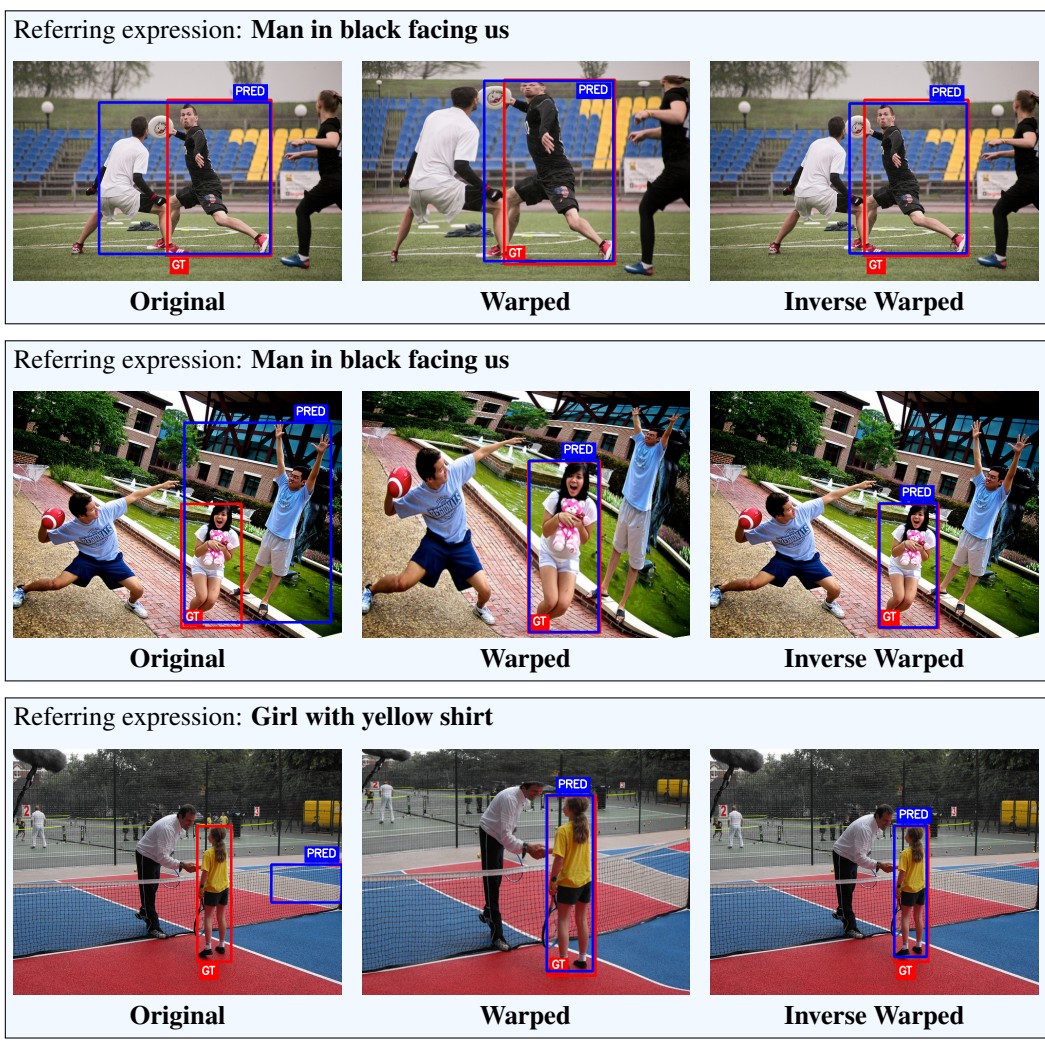

**Figure 12:** Qualitative OVOD results with LISA-LLaVA-7B. Each example shows the original, warped, and inverse-warped images. Predicted bounding boxes are in blue, ground-truth annotations in red.

### E.3 LEVERAGING SMALLER MODELS TO IMPROVE LARGER MODELS

We test whether attention maps from a smaller model can enhance the performance of a larger model. Specifically, we extract attention maps from **LLaVA-1.5-7B** to warp input images, which are then processed by the stronger **LLaVA-1.6-34B** model.

**Table 19:** TextVQA accuracy (%) of LLaVA-1.6-34B with and without image warping using attention maps from LLaVA-1.5-7B.

| Model | Without Warping | With Warping (7B maps) | $\Delta$ |
|---|---|---|---|
| LLaVA-1.6-34B | 72.6 | **74.1** | +1.5 |

This result shows that weaker models' attention maps can be repurposed to improve the performance of stronger models, highlighting a novel direction for leveraging model complementarities.

## F ATTENTION REDISTRIBUTION: REDUCTION IN MODEL UNCERTAINTY

The qualitative examples below exemplify the attention-redistributive effect of `AttWarp`. Our approach consistently improves and redirects the focus of the model toward query-specific image

regions. This can be seen by the increased coverage of relevant bounding boxes and the sharper alignment of attention peaks with target regions, in contrast to the diffused or off-target patterns often observed in the original attention maps. These visual trends support the quantitative gains reported in the main paper as discussed in Sec. 4.3 and Tab. 6d, where we showed improved localization on TEXTVQA.

**Localization precision on GQA.** To further validate query-specific focus on more datasets, we extend this study to the GQA dataset, which provides bounding-box annotations. Both Pointing Game accuracy and AM@all improve after applying `AttWarp`. Post-warp, the Pointing Game score rises from 0.412 to **0.419**, and AM@all from 0.139 to **0.154**. These gains align with the improvements observed on TEXTVQA and confirm that the rectilinear warp sharpens spatial localization across different benchmarks.

Taken together, the evidence from both TEXTVQA and GQA confirms that `AttWarp` not only preserves global distributional fidelity but also reduces model uncertainty by consistently redistributing attention mass toward query-relevant regions.

**Question:** What type of hard drive does the computer have?
**Answer:** Macintosh

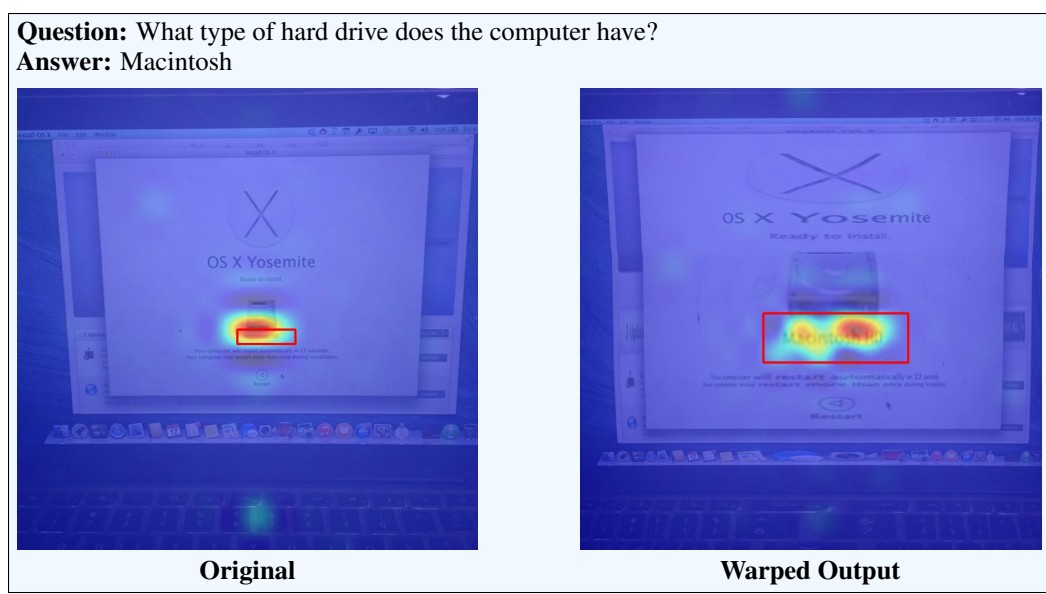

| Original | Warped Output |

**Question:** What number follows the first *fx* at the top?
**Answer:** 785

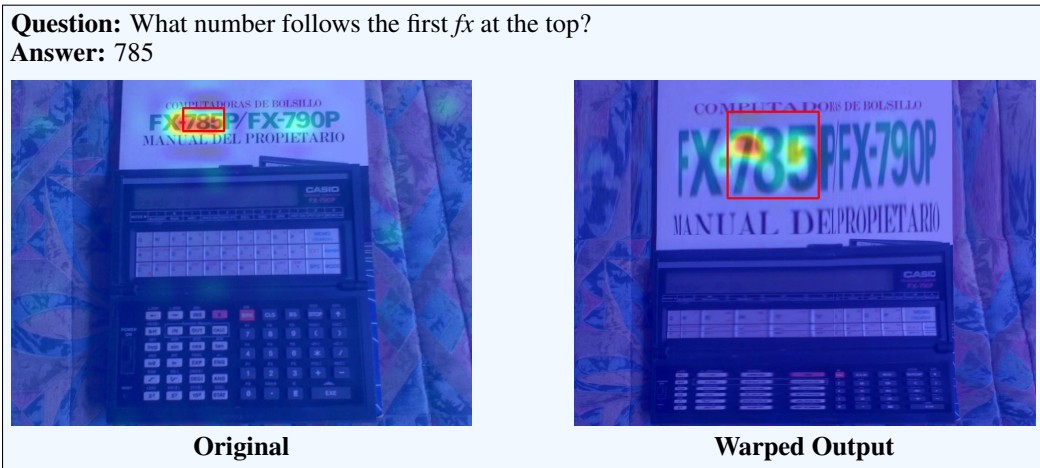

| Original | Warped Output |

**Question:** What is the alcohol content in this beer?
**Answer:** 9.5 %

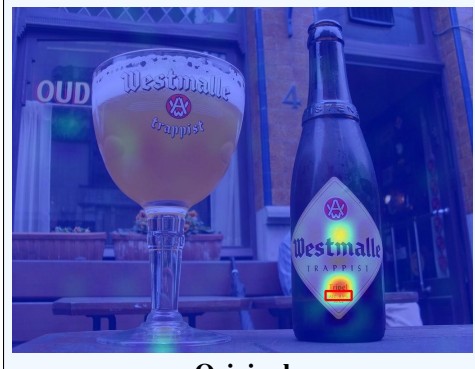 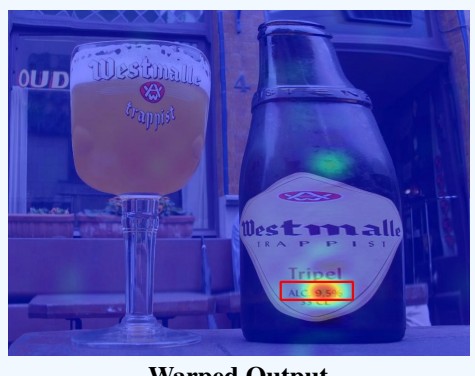

| **Original** | **Warped Output** |

## G  IMPLEMENTATION DETAILS

This section provides the implementation details for the experiments presented in Sec. 4.2 and Sec. 4.3. We first describe the setup for the quantitative results and experimental studies shown in Sec. 4.2 and Tab. 1. We then detail the configuration for the ablation studies for Sec. 4.3 (external models). We then include extended results of Tab. 1 and additional ablation analysis.

### G.1  QUANTITATIVE EVALUATION SETUP

As described in Sec. 3.4, we extract attention maps from the 20[th] layer for LLaVA-1.5-7b Liu et al. (2024b) and the 16[th] layer for Qwen2.5-VL-7b Yang et al. (2024a). The extracted attention maps are resized to the original image resolution for all downstream transformations and evaluation. For dataset-specific transforms in `AttWarp`, we apply the identity transform to TextVQA Singh et al. (2019), DocVQAMathew et al. (2021), POPELi et al. (2023), and MMMUYue et al. (2024), while for GQA Hudson & Manning (2019), the SQRT transform is used.

All experiments are conducted using 8 NVIDIA A100. For LLaVA-1.5-7b, we use the official implementation[1] with an input image resolution of $336 \times 336$ and a patch grid of $24 \times 24$. For Qwen2.5-VL-7b, all images are resized to $512 \times 512$ due to memory constraints and inference time constraints (*e.g.* ViCrop for DocVQA takes 90hours of inference time on one H100), which results in a slightly reduced baseline accuracy compared to the numbers reported in the Qwen2.5-VL-7b official release.

We set the KL divergence threshold in `AttWarp-Chain` to 0.2 (and a compulsory stopping condition of 5 iterations), based on empirical observations. For all baselines (APIYu et al. (2024), FGVPYang et al. (2023b), SoMYang et al. (2023a), and ViCropZhang et al. (2025a)), we use their official implementations. For SoM, each image is segmented with Semantic-SAM (Swin-L). For FGVP, the MLLM (LLaVA/Qwen) is first prompted to output query-specific relevant objects; then SAM (ViT-H) and CLIP (ViT-L/14) masks those regions and produce two inputs: one with the background Gaussian-blurred and one with the masked region highlighted green. Hyperparameters of FGVP and SoM are selected by grid search over 100 randomly sampled images per dataset to select the optimal values. Hyperparameters for SoM: granularity = 3, $\alpha$ = 0.4, text-size = 640; FGVP: blur $\sigma$ = 5, $\alpha$ = 0.5, IoU threshold 0.86, stability threshold 0.92, and minimum mask area 400. In the case of API and ViCrop, we exactly use the official implementation (API[2] and ViCrop[3]) with the same hyperparameters.

Accuracy is used as the evaluation metric for all datasets: for each question-answer pair $(q, a)$, the prediction is considered correct if the model output exactly matches $a$, with accuracy computed as the average over all examples. In TextVQA and DocVQA, we do not provide any OCR-extracted tokens

---

[1] `https://github.com/haotian-liu/LLaVA`

[2] `github.com/yu-rp/apiprompting`

[3] `github.com/saccharomycetes/mllms_know`

to the MLLMs—only the image and question are given, following the evaluation prompt format outlined in the respective papers. Finally, to ensure reproducibility and transparency, we will release all code, configurations, and analysis scripts required to reproduce our experiments and results.

## G.2 STABLE DIFFUSION EXPERIMENTAL DETAILS

In Appendix Sec. G.2, we use Stable Diffusion for external attention; here, we detail its experimental design. We adapt `AttWarp` to Stable Diffusion 2.1 (SD-2.1) by first inverting the input image $I$ into the model's latent space with a five-step truncated DDIM schedule [1000, 800, 600, 400, 200], conditioning on the question $q$ at every step. Each inverted latent is then forwarded through ten denoising iterations while recording the cross-attention tensors of the final two UNet layers — previously identified by Hertz et al. (2022) as exhibiting particularly sharp token-level groundings. Summing these tensors over heads and space yields token-wise energies from which we retain the top–$k$=20 tokens, average their channels into a single low-resolution attention map, apply a square-root contrast stretch, and—together with its inverse on the one-dimensional marginals—use it to drive a single `sqrt` CDF warp at $500\times500$ resolution. As reported in Table 16, this external attention improves the vanilla LLaVA baseline by **+6.7%** on TextVQA, yet remains below the gains from internal LLaVA attention (+8.8%) and from the stronger Qwen-VL (+10.0%). We attribute this gap to task mis-alignment: diffusion models are trained for literal scene synthesis (e.g. "*a giraffe in a misty forest*"), whereas visual question answering centres on self-referential queries (e.g. "*what animal is in the forest?*"), so the resulting diffusion attention maps, although sharp, do not always highlight regions most informative for answering such questions.

## G.3 TRAINING SETUP FOR `AttWarp-Distill`

We use Qwen-2.5VL-7B as the teacher model to extract attention maps and derive corresponding marginals for training the student model. Specifically, attention maps from the teacher are first extracted, after which category-specific marginals are computed for each dataset. To adaptively scale attention distributions according to task granularity, we apply distinct transformations per dataset: a square-root transform for DocVQA, a cube-root transform for GQA, and an identity transform for TextVQA and fine-grained datasets such as POPE. The student model is trained for a maximum of 20 epochs, employing early stopping based on validation-set performance. Optimization uses AdamW with a batch size of 16, learning rate of $3 \times 10^{-4}$, weight decay of $1 \times 10^{-4}$, and gradient clipping with a maximum norm of 1.0. We show the predicted marginals in Fig. 13.

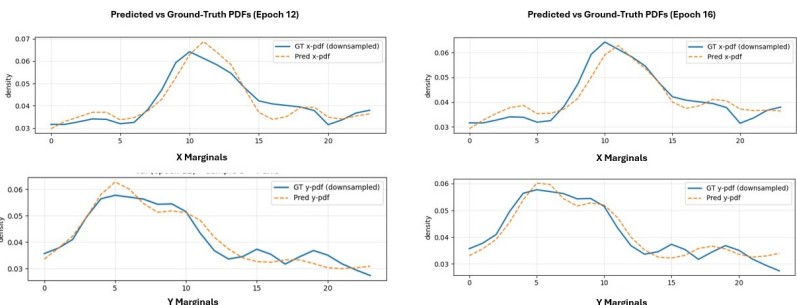

**Figure 13:** The marginal distributions predicted by the student networks closely align with the ground truth marginals, demonstrating robust knowledge transfer.

## G.4 COMPUTATIONAL COST CALCULATION AND ANALYSIS

We present a detailed cost analysis to emphasize the efficiency and practical advantages of `AttWarp` compared to the competitive baseline (ViCrop (Zhang et al., 2025a)). Here, we specifically demonstrate results for LLaVA-1.5v-7b (similar trends hold true for other models, such as Qwen). Our approach significantly surpasses the FGVP (Yang et al., 2023b), SoM (Yang et al., 2023a), and API (Yu et al., 2024) in accuracy. Therefore, we focus this analysis on ViCrop, the competitive baseline, which achieves comparable yet inferior performance to `AttWarp` across all datasets (see Sec. 4.2 and

Tab. 1). Computational complexity for all MLLMs is calculated following the methodology described in (Lin et al., 2025; Chen et al., 2023b;c; 2024a).

Standard LLaVA-1.5 processes a single $336 \times 336$ image through a ViT-L/14 encoder, producing 576 visual tokens and incurring a computational cost of 8.5 TFLOPs. In contrast, `AttWarp` requires two MLLM passes per query: the first pass extracts attention maps up to a specific layer (*e.g.*, the 20th layer of LLaVA, where we insert a hook), while the second pass generates the final output from the warped image based on these attention maps. The total cost for `AttWarp`—including both passes and the lightweight warping operation—corresponds to 1,152 vision tokens ($2 \times 576$) and 13.8 TFLOPs. The cost of the warping step itself is negligible.

In comparison, ViCrop adopts a more resource-intensive pipeline, requiring three MLLM passes: two initial passes to obtain relative attention maps (up to the 14th layer), followed by a final forward pass processing both the original and cropped image. This results in a significantly larger number of vision tokens (2,304) and a total computational load of 24.2 TFLOPs.

As summarized in Tab. 3, `AttWarp` delivers substantial computational efficiency. Specifically, it reduces the total number of vision tokens by a factor of two relative to ViCrop (1,152 *vs.* 2,304) and requires 1.5 times fewer MLLM passes (2 *vs.* 3), translating to a $1.8\times$ reduction in compute cost (13.8 TFLOPs for `AttWarp` *vs.* 24.2 TFLOPs for ViCrop per query). In terms of memory, `AttWarp` achieves a peak VRAM usage of 15.5 GB, identical to the base LLaVA-1.5v-7b model and within the capacity of standard GPUs (*e.g.*, V100s). The peak VRAM required for ViCrop is 22 GB, significantly higher (by a factor of 1.46) compared to `AttWarp`.

These results strongly motivate the use of `AttWarp`, demonstrating that it not only achieves superior accuracy (see Sec. 4.2 and Tab. 1), but also yields substantial savings in computational resources compared to approaches like ViCrop.

## H   QUALITATIVE EXAMPLES

Here, we present examples demonstrating the effectiveness of both `AttWarp` and `AttWarp-Chain`. For each case, the predicted answers are displayed beneath the corresponding images. For `AttWarp-Chain`, results are at varying depths (as expected), illustrating how our method adaptively refines warping based on the specific query and image content. This depth-wise adaptation underscores the robustness and flexibility of our approach, enabling consistently strong performance across diverse queries and visual contexts.

### H.1   ATTWARP

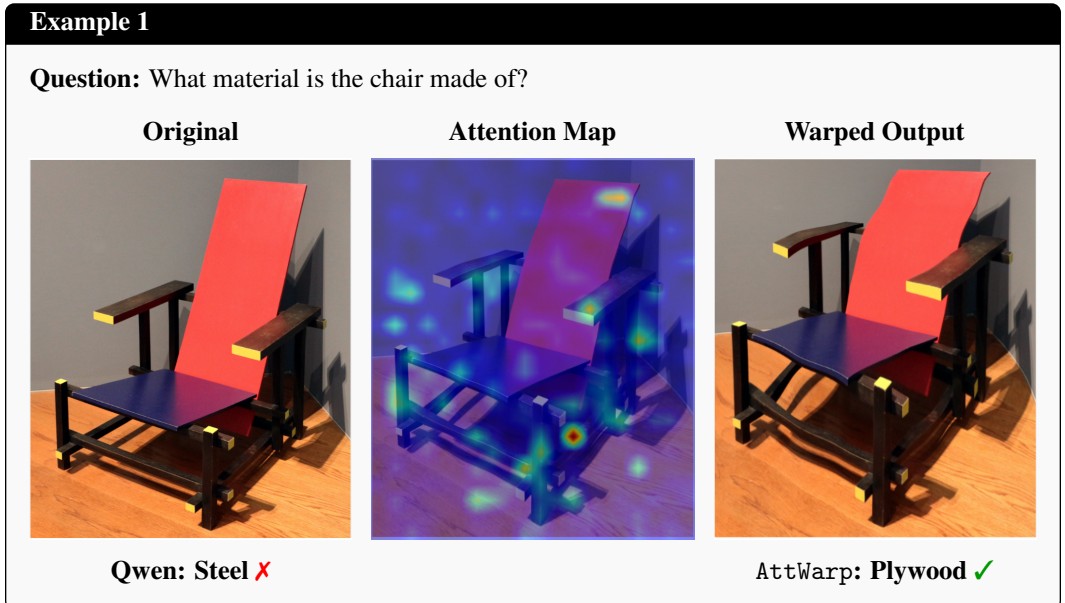

**Example 1**

**Question:** What material is the chair made of?

| Original | Attention Map | Warped Output |

**Qwen: Steel** ✗      `AttWarp`: **Plywood** ✓

**Example 2**

**Question:** Is the vehicle behind donkeys?

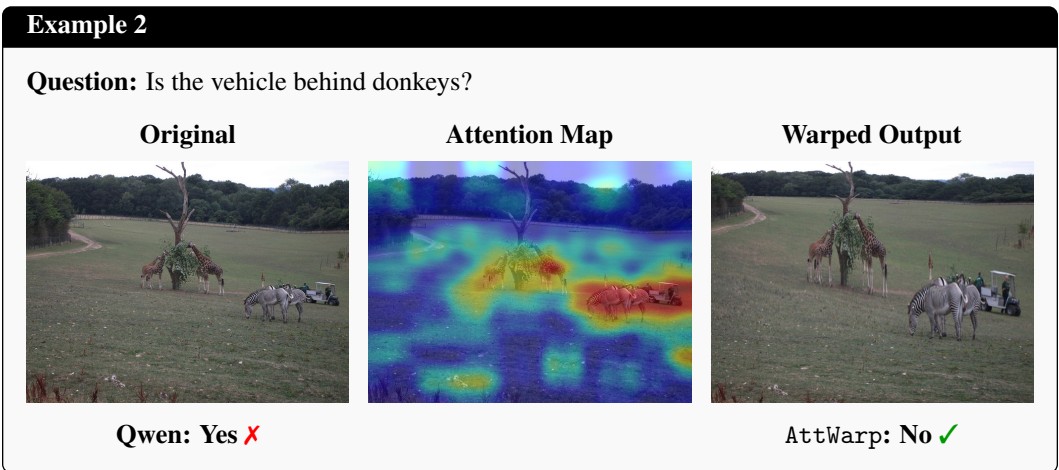

**Original**     **Attention Map**     **Warped Output**

**Qwen: Yes** ✗               `AttWarp`: **No** ✓

## H.2 `ATTWARP-CHAIN`

**Example 3**

**Question:** How many apples are there in the image?

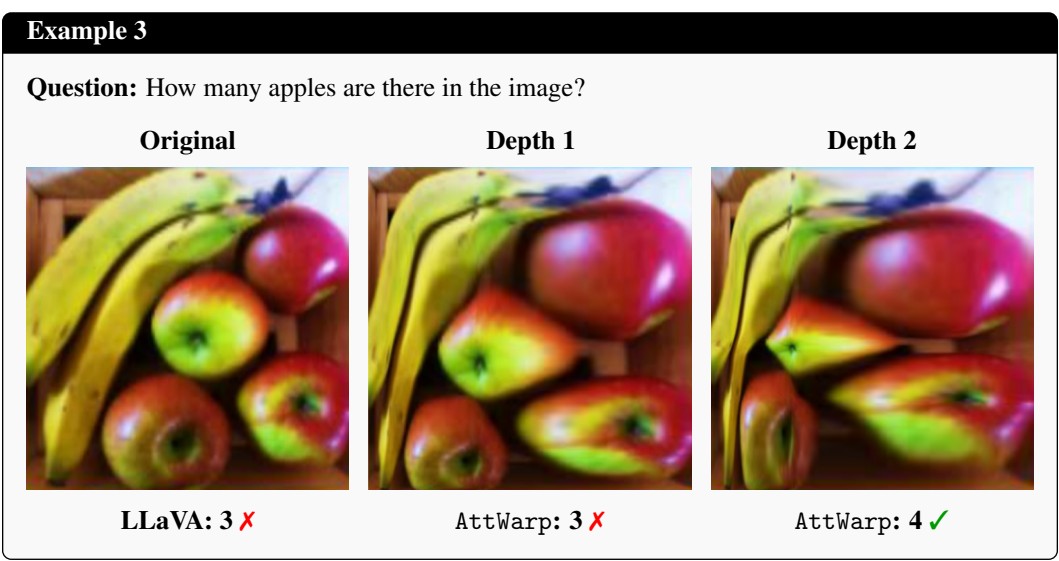

**Original**          **Depth 1**          **Depth 2**

**LLaVA: 3** ✗          `AttWarp`: **3** ✗          `AttWarp`: **4** ✓

**Example 4**

**Question:** What is a player's number?

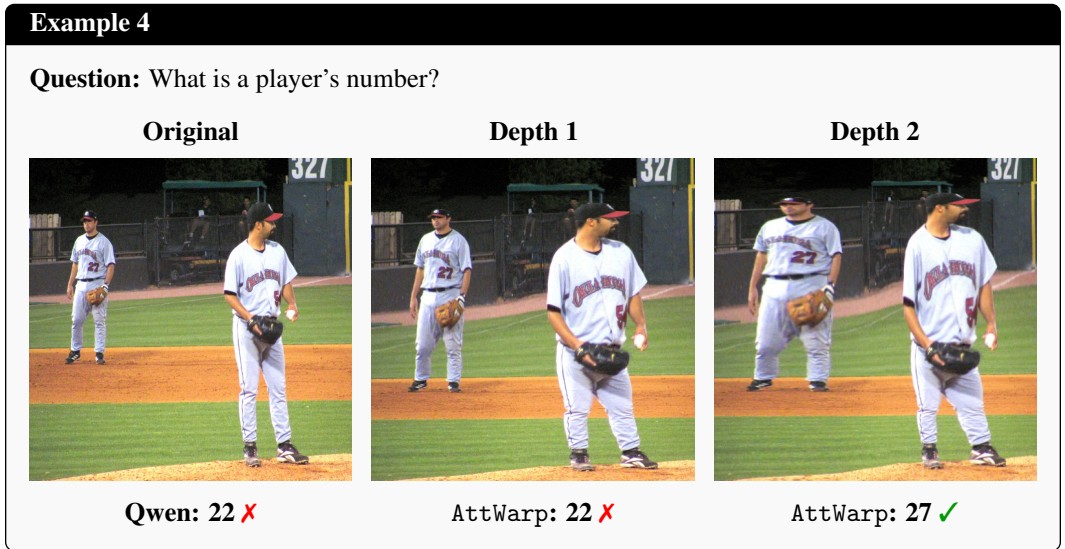

**Original**          **Depth 1**          **Depth 2**

**Qwen: 22** ✗          `AttWarp`: **22** ✗          `AttWarp`: **27** ✓

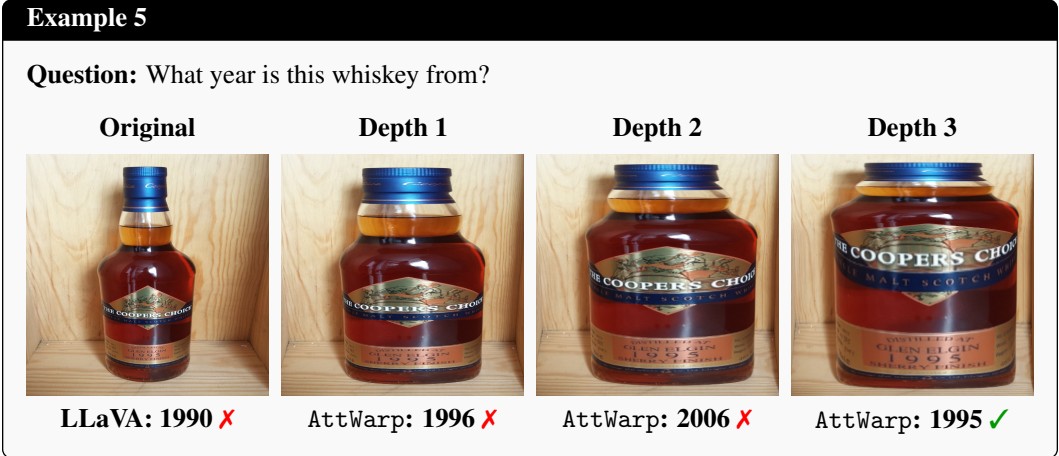

**Example 5**

**Question:** What year is this whiskey from?

| Original | Depth 1 | Depth 2 | Depth 3 |
|---|---|---|---|
| **LLaVA: 1990** ✗ | **AttWarp: 1996** ✗ | **AttWarp: 2006** ✗ | **AttWarp: 1995** ✓ |

## I  FID AND KID ANALYSIS

We quantify how closely each test variant remains on the training manifold using two complementary, CLIP-feature based distances: **Fréchet Inception Distance (FID)** and **Kernel Inception Distance (KID)**. Lower values indicate closer alignment to the training distribution.

**Why rectilinear warping preserves the distribution.** Our warp is axis-aligned and monotone,
$$(x', y') = (F_x(x), F_y(y)),$$
so its Jacobian is diagonal. At the token level this induces only per-axis rescaling of ViT patches while keeping grid orthogonality. This mirrors CLIP's *resize/crop* pre-training augmentation (e.g., RandomResizedCrop), which likewise applies axis-wise scaling before tokenization. Consequently, the pooled CLIP image embeddings for rectilinearly warped images are expected to stay on (or near) the same feature manifold as training images.

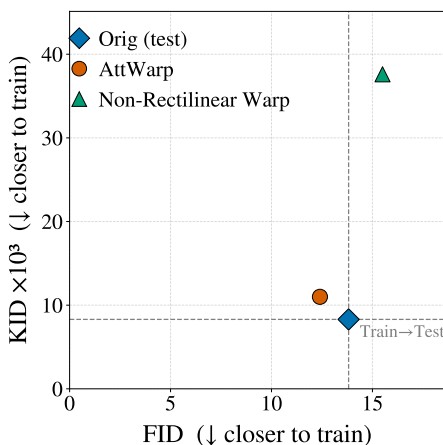

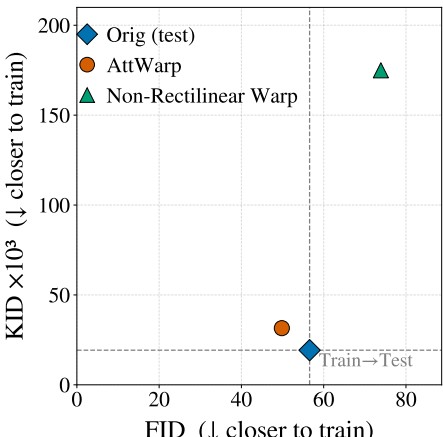

**(a)** FID–KID/$\times 10^3$ distances between the training distribution and each test variant (Original, AttWarp, Non-Rectilinear Warp) in CLIP **ViT-B/32** space. Points nearer the lower-left are closer to the training manifold.

**(b)** FID–KID/$\times 10^3$ distances between the training distribution and each test variant (Original, AttWarp, Non-Rectilinear Warp) in CLIP **ViT-L/14** space.

**Figure 14:** Comparison of Train→{Test, AttWarp, Non-Rectilinear} FID–KID across two ViT backbones.

**Setup.** We reuse 12,000 GQA-TRAIN images to define the training reference used throughout the distributional analyses (also used for the Mahalanobis study in Section 4.3). Image embeddings are obtained with CLIP **ViT-B/32** and **ViT-L/14** backbones as follows:

1. **One embedding per image.** We use CLIP's *global pooled image embedding* returned by `encode_image` (i.e., the projected [CLS] / pooled token). This is the representation CLIP trains to align with text; it is therefore the correct summary statistic for distributional tests. We do *not* mean-pool patch tokens.

2. **Reference model.** Fit a *single* full-covariance Gaussian $\mathcal{N}(\mu, \Sigma)$ on the training embeddings. A full covariance is required because CLIP feature dimensions are correlated; diagonal covariances overstate distances along correlated axes. We do *not* use a Gaussian mixture.

3. **Mahalanobis diagnostic (used for histograms referenced in Fig. 6a).** For each embedding $x$ in a split $S \in \{\text{Orig (test)}, \texttt{AttWarp}, Non-RectilinearWarp\}$, compute a single distance to the *training* Gaussian:

$$d_M(x) = \sqrt{(x - \mu)^\top \Sigma^{-1} (x - \mu)}.$$

The histogram is built from $\{d_M(x) : x \in S\}$. It is *not* a 12k-choose-2 pairwise computation. With population parameters, $d_M(x)^2$ follows $\chi_d^2$; with estimated parameters it approximates Hotelling's $T^2$.

4. **FID/KID.** Using `cleanfid` Parmar et al. (2022) with default settings (64 batch, 8 workers, CLIP 224×224 preprocessing), compute Train$\to S$ distances for each backbone. The Gaussian in step 2 is *fit once* on train; all eval splits are compared to that same reference to reveal shift (we never refit on test/warped sets).

**Results.** Fig. 14 shows Train$\to$Test distances for both backbones. In **ViT-B/32** (Fig. 14a) and **ViT-L/14** (Fig. 14b), `AttWarp` (orange circle) stays within the natural Train$\to$Test gap, reducing FID (13.8$\to$12.4 for B/32; 56.6$\to$49.8 for L/14) with only mild KID changes (8.3$\to$11.0; 19.3$\to$31.5). In contrast, the Non-Rectilinear warp (green triangle) produces a clear shift—FID rises to 15.5 and 73.9 and KID inflates markedly (37.6 and 174.9)—consistent with the Mahalanobis drift observed in Fig. 6a. These results quantitatively substantiate the design rationale above: rectilinear, axis-aligned warping preserves CLIP-feature distributional integrity, whereas free-form distortion does not.

## J    ALTERNATIVE INTERPRETATION: WARPING THE VISUAL TOKENIZATION GRID

`AttWarp` is described in the main text as a query-conditioned *image warp* followed by the model's standard visual tokenization. Concretely, from an attention score matrix we compute 1D marginal profiles and their inverse-CDF warping functions (Sec. 3.1, Eqs. (1)–(3)), and generate the warped image by inverse-warp sampling (Eq. (4)). The vision encoder then converst the warped image into patches using its usual *uniform* grid, producing a regular $h_{\text{feat}} \times w_{\text{feat}}$ lattice of visual tokens (Sec. 3.4).

**Grid view**    The same computation admits an equivalent interpretation in which we keep the original image fixed and instead *warp the patch grid*. Let the model's standard tokenization partition the (warped) image domain into $h_{\text{feat}} \times w_{\text{feat}}$ axis-aligned cells with uniform boundaries in the warped coordinate system. Mapping those cell boundaries back through the inverse warp implied by Eq. (3) yields a non-uniform, query-dependent set of cut locations in the *original* image coordinates. Because the warp is rectilinear and monotone, these mapped boundaries remain ordered and define a valid axis-aligned rectangular tiling of the original image. Thus, "warp-then-tokenize uniformly" is equivalent to "tokenize the original image with an attention-guided, non-uniform grid" (followed by the same local resampling that is already implicit in Eq. (4)). Intuitively, `AttWarp` reallocates a fixed token budget: dense (small) cells are assigned to high-attention regions, while sparse (large) cells cover low-attention areas, preserving global context since no region is discarded.

**Implication for `AttWarp-Distill`.**    In `AttWarp-Distill` (Sec. 3.3), the student is trained to predict the horizontal and vertical marginals that parameterize the same inverse-CDF warp used by `AttWarp`. Under the grid interpretation above, predicting these marginals is equivalent to predicting the *grid-cutting rule* itself: the student directly outputs a query-conditioned visual tokenization lattice (i.e., where the patch boundaries should fall, up to the model's fixed token budget). In this sense, `AttWarp-Distill` can be viewed as training an amortized, query-conditioned *visual tokenizer* (patch-grid generator) that approximates the teacher model's attention-induced token allocation, while leaving the underlying MLLM architecture and tokenizer unchanged.

