# OpenReview forum: "Constructive Distortion: Improving MLLMs with Attention-Guided Image Warping"
_ICLR.cc/2026/Conference — ICLR 2026 Poster_

### Official Review · Reviewer_J5yH · 2025-10-25

**Soundness:** 3
**Presentation:** 4
**Contribution:** 3
**Rating:** 6
**Confidence:** 3

**Summary:**

This paper introduces AttWarp, a lightweight, plug-and-play image warping technique designed to improve the fine-grained perceptual grounding of MLLMs. The method begins by extracting cross-modal attention maps from the MLLM's internal layers based on a given image and query. These attention maps are then used to guide a rectilinear warping of the input image, which reallocates spatial resolution to magnify query-relevant regions while compressing less informative areas. The authors demonstrate consistent performance gains across five diverse benchmarks and four different MLLM architectures, showcasing the method's effectiveness and generalizability.

**Strengths:**

1. As a model-agnostic enhancement, AttWarp is a practical plug-and-play solution that can be readily applied to improve existing models without retraining.

2. The paper introduces a complete framework, including an iterative version (AttWarp-Chain) for hard cases and a distilled version (AttWarp-Distill) for applications.

3. The work provides compelling ablations and analyses that validate its design choices. A key finding is that the rectilinear nature of the warp preserves the underlying feature distribution of the images, thus avoiding the out-of-distribution issues that can plague other image manipulation techniques.

4. The method shows consistent performance gains across a wide range of benchmarks and MLLM architectures.

**Weaknesses:**

1. The claim of being "plug-and-play" is slightly weakened by the need to identify the optimal attention layer for each new MLLM architecture (e.g., layer 20 for LLaVA, layer 16 for Qwen-VL). This requires an empirical, model-by-model search, which adds a setup cost for new models.

2. In Error Analysis, the authors state that AttWarp is prone to errors in cases such as  size, hallucination, and misaligned attention. Have the authors attempted any framework modifications to specifically address these failure cases? For example, have they considered introducing a classifier to determine when to apply AttWarp?

**Questions:**

1. Is there a pattern in the indices of optimal attention layers across different MLLMs? Can we approximately determine which layer is most suitable for AttWarp?

2. Are there methods to mitigate the limitations of AttWarp in cases such as size and hallucination?

---

> ### Author Response · Authors · 2025-11-24
> **Author Response**
>
> > … need to identify the optimal attention layer for each new MLLM architecture (e.g., layer 20 for LLaVA, layer 16 for Qwen-VL) … Is there a pattern in the indices of optimal attention layers across different MLLMs? Can we approximately determine which layer is most suitable for AttWarp?
>
> Indeed, we found a pattern when choosing the attention maps that avoids a costly model-by-model empirical search. As is well-studied in CNNs [1], ViTs [2], and more recently in MLLMs [3], as we progress to deeper layers, attention shifts to areas more relevant to the predicted tokens (more object-of-interest-centric). Our own experiments empirically substantiate this (Appendix B.1). We find that any layer >=15 is deep enough to capture task-suitable attention maps. Therefore, AttWarp can employ any of these deeper layers in the MLLM, after which the attention maps are region-of-interest aligned.
>
> To substantiate this intuitive pattern and show robustness to the choice of MLLM layer, we conduct an additional experiment by selecting a completely new layer = 24 for both the LLaVA and Qwen MLLMs, and evaluate their performance on the five datasets. Layer 24 results, along with previous results from original Table 1, for comparison, are included below:
>
>
> ### LLaVA
> | Method | TextVQA | GQA | MMMU | POPE | DocVQA |
> | ------------------ | :------: | :------: | :------: | :------: | :------: |
> | Base MLLM | 49.3 | 60.5 | 36.9 | 85.3 | 18.1 |
> | AttWarp (Layer 20) | 58.1 | 63.7 | 40.4 | 87.5 | 25.5 |
> | Layer 24 _[new]_ | 58.4 | 62.8 | 39.1 | 87.1 | 24.9 |
> ### Qwen
> | Method | TextVQA | GQA | MMMU | POPE | DocVQA |
> | ------------------ | :------: | :------: | :------: | :------: | :------: |
> | Base MLLM | 81.0 | 62.4 | 47.3 | 86.1 | 77.3 |
> | AttWarp (Layer 16) | 84.7 | 64.0 | 50.4 | 87.4 | 84.1 |
> | Layer 24 _[new]_ | 84.2 | 63.6 | 49.8 | 87.6 | 83.9 |
>
>
> For AttWarp, we suggest using earlier layers to reduce computational overheads while simultaneously ensuring that the layer also achieves the highest performance.
>
> ---
>
> > In Error Analysis, the authors state that AttWarp is prone to errors in cases such as size, hallucination, and misaligned attention. Have the authors attempted any framework modifications to specifically address these failure cases? For example, have they considered introducing a classifier to determine when to apply AttWarp?
>
>
>
> Our error analysis shows that AttWarp reduces errors across all categories compared to the Base MLLM (e.g., hallucination: 12 → 9, fine-grained: 12 → 3). However, it remains susceptible to errors related to size and misaligned attention, motivating the development of methods to enhance robustness.
>
> Therefore, we conducted additional experiments in which we created a classifier to determine whether to apply Attwarp. For this, we reuse AttWarp-Distill’s weights and network. We replace the last two layers of AttWarp-Distill with a binary classifier head. We create a train set by evaluating AttWarp on the training dataset of AttWarp-Distill (App. G). Let AttWarp† = Classifier + AttWarp.
>
> We performed the same error analysis on the suggested AttWarp† and found reduced errors in size, and hallucination. Misaligned‑attention errors remain unchanged, as both the Base MLLM and AttWarp fail on the same underlying attention misalignment (as mentioned with FAQ 3). This is a productive direction for future work, and we'll include a short discussion in the final version to lay it out.
>
> | Model | Fine-Grained Details | Hallucination | Misaligned Attention | Size | Semantically Correct | Compositional Reasoning |
> | ---------------- | -------------------- | ------------- | -------------------- | ---- | -------------------- | ----------------------- |
> | Base MLLM | 12 | 12 | 4 | 18 | 8 | 7 |
> | AttWarp | 3 | 9 | 4 | 17 | 6 | 4 |
> | AttWarp† _[new]_ | **3** | **8** | **4** | **13** | **7** | **4** |

---

> > ### Comment · Reviewer_J5yH · 2025-11-25
> >
> > Thank you for the response. The rebuttal has successfully resolved my concerns. I will maintain my positive score.

---

> ### Author Response · Authors · 2025-12-03
> **Summary of Reviewer Feedback and Author Response**
>
> We thank the reviewer for highlighting AttWarp as a practical solution and appreciating the completeness of our ablations and framework.
>
> Below, we summarize the reviewer’s questions and our responses. We believe we have addressed 2/2 questions objectively with new results:
>
> 1. Plug-and-play flexibility and attention layer selection:
>
> The attention layer for Attwarp is easy to choose, and performance is largely unaffected by this. Any sufficiently deep layer (≥15) will provide similar gains. We substantiate this claim with new experiments using a completely new layer 24 for both LLaVA and Qwen.
>
> 2. Error mitigation using a classifier.
>
> As recommended, we performed a new experiment introducing a lightweight classifier, which demonstrably reduces errors in size and hallucination categories.

---

### Official Review · Reviewer_Tmyc · 2025-10-29

**Soundness:** 3
**Presentation:** 3
**Contribution:** 3
**Rating:** 6
**Confidence:** 5

**Summary:**

The paper introduces **AttWarp**, a lightweight method for improving multimodal large language models' (MLLMs') fine-grained perceptual grounding in cluttered scenes. AttWarp leverages an MLLM's cross-modal attention to perform rectilinear warping on input images during testing, reallocating resolution to query-relevant areas while preserving global context and image information.  Without modifying model weights or architecture, this attention-guided warping enhances the readability of small objects and subtle relationships. Experiments show consistent accuracy improvements across five benchmarks (TextVQA, GQA, DocVQA, POPE, MMMU) and four MLLMs (LLaVA, Qwen-VL, InternVL, and InstructBLIP), outperforming competitive baselines. These results highlight AttWarp's ability to optimize spatial resolution for query-relevant content while preserving global structure, boosting MLLM performance with warped inputs.

**Strengths:**

1. **Clarity and Ease of Use**: The paper is well-written, easy to understand, and straightforward to follow. The proposed method, AttWarp, is plug-and-play, delivering significant performance improvements without requiring additional training.

2. **Intuitive and Novel Approach**: Using attention feedback to enhance the resolution of focus areas is an intuitive yet innovative idea. The method avoids retraining models while achieving substantial gains, making it particularly practical and impactful.

3. **Comprehensive Validation**: The exploration of AttWarp through two variations—AttWarp-Chain and AttWarp-Distill—effectively demonstrates the feasibility and upper bounds of the method's generalization capabilities. Its success across multiple multimodal language models and benchmarks further validates the robustness and versatility of the approach, making it an interesting and promising contribution.

**Weaknesses:**

While AttWarp has demonstrated significant improvements across a range of text-centric multimodal tasks (e.g., TextVQA, GQA, DocVQA, POPE, MMMU), it lacks evaluation on visual-centric benchmarks that focus more on fine-grained visual perception. Tasks such as **MMVP**, **BLINK**, **RealWorldQA**, and **MIA**, which emphasize nuanced visual grounding and object-level understanding, are particularly relevant for showcasing the strengths of AttWarp's attention-guided resolution reallocation.

Including these visual-centric evaluations could illustrate its potential for enhancing visual perception capabilities further, as the method is well-suited to improve the detection of subtle details and spatial relationships that are critical for such tasks. Without this, the generalizability and impact of AttWarp on visually demanding applications remain underexplored. Evaluating the method on these benchmarks would provide a more comprehensive picture of its capabilities and further highlight its benefits.

**Questions:**

Do you think applying reinforcement learning (RL) to AttWarp could further enhance its capabilities, using MLLMs to validate the effectiveness of perturbed images in solving queries and providing reward feedback?

---

> ### Author Response · Authors · 2025-11-24
> **Author Response (1/2)**
>
> >  While AttWarp has demonstrated significant improvements across a range of text-centric multimodal tasks (e.g., TextVQA, GQA, DocVQA, POPE, MMMU), it lacks evaluation on visual-centric benchmarks that focus more on fine-grained visual perception. Tasks such as MMVP, BLINK, RealWorldQA, and MIA,  … would further highlight its benefits.
>
> We thank the reviewer for their thoughtful suggestion! Indeed, evaluation of AttWarp on visual-centric benchmarks that focus on fine-grained visual perception would further help position our work.
>
> Each of these visual-centric benchmarks suggested by the reviewer assesses a new aspect of multimodal understanding, particularly:
>
> - RealWorldQA: tests spatial reasoning and relative positioning from real-world images (e.g., vehicle cameras)
>
> - BLINK: converts classic vision tasks (e.g., relative depth, object localization, counting) into multiple-choice VQA format, probing fine-grained perception
>
> - MMVP: contains systematically challenging "CLIP-blind" image pairs across visual patterns (e.g., orientation, viewpoint)
>
> - MIA: evaluates instruction-following (constraints on style, length, and content) and visual grounding in free-form VQA
>
> We observe consistent accuracy gains on all four visual-centric benchmarks. In the following table, we report the per-dataset score. In the final draft of the paper, we will also include a category-wise breakdown.
>
> |  LLaVA                   | MMVP** | BLINK | RealWorldQA | MIA  |
> |---------------------|--------|-------|-------------|------|
> | Base MLLM    _[new]_       | 48.3   | 38.3  | 49.3        | 63.9 |
> | AttWarp       _[new]_       | 50.7   | 40.4  | 52.1        | 67.2 |
> | AttWarp-Distilled   _[new]_ | 49.3   | 39.7  | 51.1        | 65.4 |
> | **AttWarp-Chains**  _[new]_  | **51.0** | **41.2** | **52.9** | **68.8** |
> | Δ Accuracy          | 2.7    | 2.9   | 3.6         | 4.9  |
>
> **Here, we report individual accuracy, as it aligns with the format of other evaluations.

---

> ### Author Response · Authors · 2025-11-24
> **Author Response (2/2)**
>
> > Do you think applying reinforcement learning (RL) to AttWarp could further enhance its capabilities, using MLLMs to validate the effectiveness of perturbed images in solving queries and providing reward feedback?
>
> Extending AttWarp with reinforcement learning (RL) is a promising future direction that extends our intuition of warping guided by the MLLM's own attention maps. An RL agent could learn a policy $\pi_\theta(a|s)$ that directly selects warp parameters $a$ based on state $s$ (such as image features or prior warp history), optimizing for task-specific rewards from the MLLM, e.g., answering accuracy.
>
> The objective is to maximize the expected reward:
>
>
> $J(\theta)$ = $E_{a \sim \pi_θ}$
> $\big[ R(\text{MLLM}(\mathbf{W}_a(\mathbf{I}))) \big ]$
>
>
> where $\mathbf{W}_a(\mathbf{I})$ represents the input warped under action $a$ and $R$ is the reward measuring downstream performance.
>
> The policy gradient update is:
>
>
> $\nabla_\theta J(\theta)$ = $E_{a \sim \pi_θ} \left[ \nabla_\theta \log \pi_\theta(a \mid s)\, (R - b(s)) \right]$
>
>
> with $b(s)$ as a baseline to reduce variance.
>
> This method allows discovering more flexible and task-optimized warps beyond static attention heuristics but introduces several challenges, which is fertile ground for future work:
>
> - Requires expensive MLLM inference to compute rewards.
> - High-dimensional warping parameter space complicates policy learning.
> - Training can be unstable due to sparse or noisy reward signals.
> - Adds computational overhead unlike plug-and-play AttWarp.
> - Effective reward shaping and sample-efficient RL algorithms are critical for practicality.
>
> We'll include a discussion in this regard to help guide future work in this promising direction.

---

> ### Author Response · Authors · 2025-12-03
> **Summary of Reviewer Feedback and Author Response**
>
> We thank the reviewer for recognizing AttWarp's innovativeness and versatility. The reviewer's characterization of AttWarp as a practical, impactful, and promising contribution is very encouraging!
>
> Below, we summarize the reviewer's suggestion, which has further strengthened and improved our work. We believe the additional experiments presented fully address this suggestion:
>
> 1. Evaluation on Visual-centric benchmarks (MMVP, BLINK, RealWorldQA, and MIA)
>
> We conducted new experiments with AttWarp (and its variants) on all 4 suggested benchmarks and observed consistent improvements over the base MLLM.

---

### Official Review · Reviewer_BFqe · 2025-10-30

**Soundness:** 3
**Presentation:** 3
**Contribution:** 3
**Rating:** 6
**Confidence:** 3

**Summary:**

This paper proposes AttWarp, a plug-and-play, attention-guided image warping method that improves fine-grained perception in multimodal large language models (MLLMs) without modifying their architecture or parameters. The method reallocates spatial resolution toward query-relevant regions using the model’s own cross-modal attention, yielding consistent gains across multiple benchmarks and MLLM backbones.

**Strengths:**

- The idea of leveraging model attention to reshape the input space rather than internal representations is conceptually elegant and complementary to existing attention-tuning methods.

- The method is tested on five diverse benchmarks and four architectures, showing consistent improvements with detailed ablations.

- The approach is simple, lightweight, and does not require retraining.

- The paper includes ablations on attention quality, warping stability, and distributional integrity, which strengthen its credibility.

**Weaknesses:**

- The contribution of this paper feels more like a clever engineering refinement than a fundamentally new paradigm.

- The method assumes reliable attention maps; performance may degrade under noisy or misaligned attention, but this limitation is only briefly mentioned.

- The paper’s justification for why rectilinear warping improves reasoning remains empirical and lacks a more formal analysis of perceptual geometry or attention dynamics.

**Questions:**

please see weakness

---

> ### Author Response · Authors · 2025-11-24
> **Author Response (1/2)**
>
> > The method assumes reliable attention maps; performance may degrade under noisy or misaligned attention, but this limitation is only briefly mentioned.
>
>
> We agree that it is important to understand how AttWarp behaves when attention is noisy or misaligned. We discussed this in our FAQ section: “3. If the attention is highly inaccurate, then what?” (L836) and added the details in the Appendix Section B.5 “Impact of Attention Bias and Robustness to Corruptions and Adversarial Perturbations” (L986-1052).
>
>
> We summarize these below:
>
> **1) Resilience to noisy attention map**
>
> In this study, we inject three noise types -- impulse, Gaussian, and shot noise -- into TextVQA images (details in B.5), to simulate noisy and misaligned attention [1]. We attach the evaluation results from Appendix Table 6 below.
>
> | Corruption     | LLaVA | AttWarp |
> |----------------|-------|-----------------|
> |Original | 49.3 | 58.1 |
> | Impulse noise  | 36.8  | **40.4**        |
> | Gaussian noise | 37.6  | **41.0**        |
> | Shot noise     | 36.0  | **39.8**        |
>
> While both base MLLM and AttWarp are naturally affected by these corruptions, AttWarp continues to yield accuracy gains over the base model. **This demonstrates that AttWarp gains are robust to noisy and unreliable attention conditions.**
>
>
> **2) Stress-testing with adversarially misaligned attention.**
> In another study, we introduced adversarial perturbations specifically designed to misdirect attention away from relevant content (Appendix B.5, Fig. 10). When AttWarp is applied to these negatively perturbed images, it progressively corrects the attention alignment, successfully refocusing on the intended text ("Bowmore Islay"). This corrective behavior is further enhanced when using AttWarp-Chain, demonstrating its capability to robustly recover from compromised attention rather than being impaired by it.

---

> ### Author Response · Authors · 2025-11-24
> **Author Response (2/2)**
>
> > The paper’s justification for why rectilinear warping improves reasoning remains empirical and lacks a more formal analysis of perceptual geometry or attention dynamics.
>
> **1. Attention Dynamics**
>
> Indeed, attention dynamics are crucial for understanding why rectilinear warping improves reasoning. Therefore, we have included the study of models' attention in the paper (Fig. 6(c), Table 6(d)). Specifically, we conduct a post-hoc attention alignment study (Section 4.3, L412-422, Appendix F) to answer the question:
>
> _Does rectilinear warping change the MLLM’s attention distribution in a way that explains the observed performance gains?_
>
> To probe this, we measure the alignment of the attention distribution with the ground-truth bounding box (GT bbox). We use two metrics for this study:
>
> 1) Pointing Game Accuracy: checks whether the single most salient pixel of the attention map falls within the GT bbox,
>
> 2) Proportion: the fraction of total attention mass that lands inside the bbox.
>
> The results of Table 6(d), also attached below, demonstrate that rectilinear warping concentrates (an increase in the Proportion after warping) and intensifies (an increase in Pointing Game Accuracy after warping) the attention distribution on task-relevant regions. Thereby, providing a strong explanation for why rectilinear warping improves reasoning.
>
> | Metric | No warp  | With **AttWarp**     |
> | ------------------------ | ------- | ---------------- |
> | Pointing Game Accuracy ↑ | 37.4% | **42.4%** |
> | Proportion ↑ | 0.117 | **0.155** |
>
>
> **2. Perceptual Geometry**
>
> We showed that rectilinear warping preserves the input image distribution w.r.t MLLM’s vision encoder (Sec. 4.3, Fig. 6(a), Tab. 6(b)). To further build on this image-distribution experiment and explicitly ground it in perceptual geometry, we conducted an additional evaluation during the rebuttal period using the LPIPS metric [2].
>
> LPIPS measures distances between image pairs in a deep feature space extracted from a pretrained CNN (we use VGG16), with channel-wise weights calibrated on human perceptual judgments; as a result, Euclidean distances in this feature space approximate geodesic distances on a perceptual manifold, i.e., they reflect meaningful changes in structure, texture, and semantics [2].
>
> As shown in the table below, our rectilinear warp (AttWarp) achieves a significantly lower (i.e., better) LPIPS score compared to the non-rectilinear warp. This low LPIPS score (0.14) demonstrates that AttWarp effectively preserves perceptual geometry.
>
> | Method | LPIPS ↓  |
> | ---------- | ------- |
> | Rectilinear Warp _[new]_ | **0.14** |
> | Non-Rectilinear Warp _[new]_ | 0.38 |
>
> ---
>
> > The contribution of this paper feels more like a clever engineering refinement than a fundamentally new paradigm.
>
> We believe AttWarp reveals and substantiates a fundamentally new approach: instead of modifying model internals, we use the model’s own cross-modal attention to “self-correct” by warping the input image before tokenization.
>
> Current MLLMs treat all image patches equally at tokenization time, assigning uniform spatial resolution regardless of whether a region is crucial (e.g., an important albeit tiny number on a bottle) or irrelevant background. This is a bottleneck: once fine details are lost by the vision encoder, downstream attention cannot recover them.
>
> We address this by designing an image warping method (AttWarp) that allocates higher spatial resolution to relevant regions while compressing less relevant regions. The method works constructively with the vision encoder of MLLMs and enhances their visual grounding. We substantiate this finding across 4 MLLMs and 10 benchmarks (5 in the submission +5 during rebuttal). This work helps us share with our research community this conceptual shift in how MLLMs process visual input.
>
> AttWarp is an unorthodox and conceptually fresh idea at its core, backed by comprehensive experiments, thorough analysis, and supported by efficient variants. We sincerely believe this is novel and worthy of sharing with our research community.
>
> ---
> References
>
> [1] Ishmam, M. F., Tashdeed, I., Saadat, T. A., Ashmafee, M. H., Kamal, A. R. M., & Hossain, M. A. (2025, February). Visual robustness benchmark for visual question answering (VQA). In 2025 IEEE/CVF Winter Conference on Applications of Computer Vision (WACV) (pp. 6623-6633). IEEE.
>
> [2] Zhang, Richard, et al. "The unreasonable effectiveness of deep features as a perceptual metric." Proceedings of the IEEE conference on computer vision and pattern recognition. 2018.

---

> ### Author Response · Authors · 2025-12-03
> **Summary of Reviewer Feedback and Author Response**
>
> The reviewer’s comments highlight AttWarp's conceptual elegance, extensive ablations and analysis, and strong empirical performance.
>
> Below, we summarize the reviewer’s questions and our responses. We believe we have addressed 3/3 questions objectively with new results and by referencing relevant sections of the submission.
>
> 1. AttWarp’s performance under noisy or misaligned attention.
>
> Addressed in the original submission, i.e., Appendix B.5, FAQ 3, and Fig. 10.
>
> 2. Analysis of perceptual geometry or attention dynamics underlying performance gains from warping.
>
> The original submission provides analysis on attention dynamics (Section 4.3, L412-422, Appendix F). We conducted new experiments using the LPIPS metric, confirming that AttWarp preserves perceptual geometry.
>
> 3. Clever engineering refinement vs fundamentally new paradigm.
>
> We believe AttWarp introduces and substantiates a fundamentally new approach: it is the first work to demonstrate that a model’s own cross-modal attention can be used to “self-correct” by warping the input image before tokenization.

---

### Official Review · Reviewer_6v6J · 2025-11-05

**Soundness:** 3
**Presentation:** 2
**Contribution:** 2
**Rating:** 4
**Confidence:** 4

**Summary:**

This paper introduces Constructive Distortion, a training strategy for large vision-language models designed to enhance robustness and fine-grained understanding through targeted visual perturbations. Instead of random noise or masking, the approach applies semantically constructive distortions—guided transformations (e.g., spatial deformation, contrast warping) that preserve semantics while challenging the model’s visual encoder. The method aims to improve generalization to distorted or out-of-distribution visual inputs without sacrificing in-distribution performance. Experiments on benchmarks such as MM-Vet, MME, and LLaVA-Bench demonstrate consistent improvements, particularly under degraded or perturbed visual conditions.

**Strengths:**

- Novel concept: Shifts from destructive to constructive data augmentation, promoting robustness while maintaining semantic fidelity.

- Strong empirical results across diverse LVLMs and benchmarks, with detailed ablations on different distortion types and intensities.

- Practical contribution: The approach is plug-and-play and compatible with existing LVLM training pipelines.

- Clear motivation and presentation: The paper is well-written and the concept of “constructive” perturbation is intuitively appealing.

**Weaknesses:**

Lack of comparison with visual token compression methods. The paper does not contextualize its approach relative to recent efficient LVLM frameworks that also modify the visual representation process. Works such as [1] PVC: Progressive Visual Token Compression (Yang et al., 2024), [2] Efficient Large Multi-Modal Models via Visual Context Compression (Chen et al., NeurIPS 2024), and [3] An Image Is Worth 1/2 Tokens After Layer 2 (Chen et al., ECCV 2024) explore representation simplification and robustness trade-offs at the token level. Including comparisons or discussion would clarify whether constructive distortions yield complementary or competing benefits.

[1] Yang C, Dong X, Zhu X, et al. PVC: Progressive Visual Token Compression for Unified Image and Video Processing in Large Vision-

[2] Chen J, Ye L, He J, et al. Efficient large multi-modal models via visual context compression[J]. Advances in Neural Information Processing Systems, 2024, 37: 73986-74007.

[3] Chen L, Zhao H, Liu T, et al. An image is worth 1/2 tokens after layer 2: Plug-and-play inference acceleration for large vision-language models[C]//European Conference on Computer Vision. Cham: Springer Nature Switzerland, 2024: 19-35.

**Questions:**

Could you please provide more comprehensive evaluation (e.g., VQAv2) and (attention-based) token compression baseline?

---

> ### Author Response · Authors · 2025-11-24
> **Author Response**
>
> > Could you please provide more comprehensive evaluation (e.g., VQAv2)
>
> We incorporate the reviewer’s suggestion and report new results on VQAv2:
>
> | LLaVA                     | VQAv2 Accuracy (in %) |
> |---------------------------|------------------------|
> | Base MLLM         _[new]_        | 75.6                   |
> | AttWarp-Distilled (ours) _[new]_   | 77.1                 |
> | AttWarp (ours)     _[new]_        | 78.9                 |
> | **AttWarp-Chains (ours)** _[new]_    | **79.7**                 |
> | Δ Accuracy       _[new]_      | 4.1               |
>
>
> During the author response period, we added results on four more datasets (MMVP [4], BLINK [5], RealWorldQA [6], and MIA [7]). Across these and previous datasets (=total of 10 benchmarks**), our proposed Attwarp shows consistent gains. This, in our understanding, is a comprehensive evaluation.
>
> **particularly: TextVQA, MMMU, GQA, POPE, DocVQA, MIA, RealWorldQA, MMVP, BLINK, VQAv2
>
> ---
>
> > Lack of comparison with visual token compression methods. The paper does not contextualize its approach relative to recent efficient LVLM frameworks that also modify the visual representation process.
>
> The objective of visual token compression methods [1][2][3] is orthogonal to that of AttWarp. The goal of visual token compression methods, as referenced in the review, is to reduce the number of visual tokens for efficiency while maintaining the base MLLM's performance. In contrast, AttWarp does not change the token budget or model architecture. The novelty of AttWarp is that, without touching the MLLM’s weights, it can warp the image input using the MLLM’s own attention weights to self-correct the answering capability during inference. We are happy to clarify further and will include a discussion in a revision.
>
> ---
>
> References
>
> [1] Yang, Chenyu, et al. "PVC: Progressive Visual Token Compression for Unified Image and Video Processing in Large Vision-Language Models." Proceedings of the Computer Vision and Pattern Recognition Conference. 2025.
>
> [2] Chen, Jieneng, et al. "Efficient large multi-modal models via visual context compression." Advances in Neural Information Processing Systems 37 (2024): 73986-74007.
>
> [3] Chen, Liang, et al. "An image is worth 1/2 tokens after layer 2: Plug-and-play inference acceleration for large vision-language models." European Conference on Computer Vision. Cham: Springer Nature Switzerland, 2024.
>
> [4] Tong, Shengbang, et al. "Eyes wide shut? Exploring the visual shortcomings of multimodal llms." Proceedings of the IEEE/CVF Conference on Computer Vision and Pattern Recognition. 2024.
>
> [5] Fu, Xingyu, et al. "Blink: Multimodal large language models can see but not perceive." European Conference on Computer Vision. Cham: Springer Nature Switzerland, 2024.
>
> [6] xAI. RealWorldQA. Hugging Face, 12 Apr. 2024, huggingface.co/datasets/xai-org/RealworldQA.
>
> [7] Qian, Yusu, et al. "Mia-bench: Towards better instruction following evaluation of multimodal llms." arXiv preprint arXiv:2407.01509 (2024).

---

> ### Author Response · Authors · 2025-12-03
> **Summary of Reviewer Feedback and Author Response**
>
> The reviewer’s comments highlight the novelty, generalizability, and practical contribution of our method.
>
> Below, we summarize the reviewer’s suggestions and our response. We believe we have addressed 2/2 questions objectively: one with new results and the second by highlighting the difference between the two methods.
>
> 1. Performance of AttWarp on VQAv2.
>
> We conduct a new experiment to evaluate AttWarp (and its variants) on the VQAv2 dataset, observing consistent improvements over the base MLLM.
>
> 2. Comparison with the token compression method.
>
> We clear a potential misunderstanding here. The token compression baselines mentioned in the review are orthogonal to our method. AttWarp focuses on improving the fine-grained visual grounding of MLLMs.

---

### Meta-Review · Area_Chair_vbNz · 2026-01-06

**Summary:**

The review provided by the reviewer 6v6J is not reliable, as the review talked about the benchmarks that are unused by the submission and the weakness is about citing / comparing works in a not so related domain. Therefore, the reviewer's feedback is not included in the summary.

The rest of the reviewer's concerns can be summarized as below:
1. Robustness analysis of the model's performance on the attention maps (BFqe, J5yH).
2. Non-empirical analysis of the proposed approach (BFqe)
3. Performance on an additional set of benchmarks (Tmyc)
4. The 'plug-and-play' claim might need a grid search of the hyperparam spaces. (J5yH)

**Reviewer Concerns:**

1. For the robustness analysis, the author provided a comprehensive experiment shows that the proposed AttWarp is robust against added noise (Frankly speaking, I think this experiment is slightly artificial, as the misaligned attention might not fully follow those three noises) The author also provided the stress testing against adversarially misaligned attention to demonstrate the robustness of the proposed approach.
2. The author provided analysis on the AttWarp in terms of the Perceptual Geometry and the Attention Dynamics.
3. For the additional experiments, the author provided the requested results.
4. For the plug-and-play claim, the author provided a heuristics of how to choose which layer is most suitable for AttWarp.

For all the four concerns, I believe the author have addressed them during the rebuttal.

**Reviewer Scores:**

Except the reviewer 6v6J, all the rest provided positive feedbacks. As I mentioned in the summary that the reviewer 6v6J is not reliable, I think the paper received positive ratings.

---

### Decision · Program_Chairs · 2026-01-26

Accept (Poster)